# Eleven genomic loci affect plasma levels of chronic inflammation marker soluble urokinase-type plasminogen activator receptor

Joseph Dowsett [1✉], Egil Ferkingstad [2], Line Jee Hartmann Rasmussen[3,4], Lise Wegner Thørner[1], Magnús K. Magnússon [2,5], Karen Sugden[3], Gudmar Thorleifsson[2], Mike Frigge [2], Kristoffer Sølvsten Burgdorf[1], Sisse Rye Ostrowski [1], Erik Sørensen[1], Christian Erikstrup [6], Ole Birger Pedersen [7], Thomas Folkmann Hansen [8,9], Karina Banasik [9], Søren Brunak [9], DBDS Genomic Consortium*, Vinicius Tragante [2,10], Sigrun Helga Lund [2], Lilja Stefansdottir[2], Bjarni Gunnarson[2], Richie Poulton [11], Louise Arseneault [12], Avshalom Caspi[3,12,13,14], Terrie E. Moffitt[3,12,13,14], Daníel Gudbjartsson [2,15], Jesper Eugen-Olsen[4], Hreinn Stefánsson[2], Kári Stefánsson [2,5] & Henrik Ullum[1]

Soluble urokinase-type plasminogen activator receptor (suPAR) is a chronic inflammation marker associated with the development of a range of diseases, including cancer and cardiovascular disease. The genetics of suPAR remain unexplored but may shed light on the biology of the marker and its connection to outcomes. We report a heritability estimate of 60% for the variation in suPAR and performed a genome-wide association meta-analysis on suPAR levels measured in Iceland ($N = 35,559$) and in Denmark ($N = 12,177$). We identified 13 independently genome-wide significant sequence variants associated with suPAR across 11 distinct loci. Associated variants were found in and around genes encoding uPAR (*PLAUR*), its ligand uPA (*PLAU*), the kidney-disease-associated gene *PLA2R1* as well as genes with relations to glycosylation, glycoprotein biosynthesis, and the immune response. These findings provide new insight into the causes of variation in suPAR plasma levels, which may clarify suPAR's potential role in associated diseases, as well as the underlying mechanisms that give suPAR its prognostic value as a unique marker of chronic inflammation.

[1] Department of Clinical Immunology, Copenhagen University Hospital, Copenhagen, Denmark. [2] deCODE genetics/Amgen Inc., Reykjavik, Iceland. [3] Department of Psychology and Neuroscience, Duke University, Durham, NC, USA. [4] Department of Clinical Research, Copenhagen University Hospital Amager and Hvidovre, Hvidovre, Denmark. [5] Faculty of Medicine, School of Health Sciences, University of Iceland, Reykjavik, Iceland. [6] Department of Immunology, Aarhus University Hospital, Aarhus, Denmark. [7] Department of Immunology, Naestved Hospital, Naestved, Denmark. [8] Danish Headache Center, Copenhagen University Hospital, Rigshospitalet Glostrup, Glostrup, Denmark. [9] Novo Nordisk Foundation Center for Protein Research, Faculty of Health and Medical Sciences, University of Copenhagen, Copenhagen, Denmark. [10] Department of Cardiology, Division Heart and Lungs, UMC Utrecht, University of Utrecht, Utrecht, Netherlands. [11] Dunedin Multidisciplinary Health and Development Research Unit, Department of Psychology, University of Otago, Dunedin, New Zealand. [12] Social, Genetic, and Developmental Psychiatry Centre, Institute of Psychiatry, Psychology, and Neuroscience, King's College London, London, UK. [13] Department of Psychiatry and Behavioral Sciences, Duke University School of Medicine, Durham, NC, USA. [14] Center for Genomic and Computational Biology, Duke University, Durham, NC, USA. [15] School of Engineering and Natural Sciences, University of Iceland, Reykjavik, Iceland. *A list of authors and their affiliations appears at the end of the paper. ✉email: joseph.dowsett@regionh.dk

The plasma protein soluble urokinase-type plasminogen activator receptor (suPAR) is a non-specific biomarker for chronic inflammation (also termed low-grade inflammation) and was recently identified as a key molecule of senescent cells[1]. It structurally consists of three domains ($D_I–D_{III}$) and is the soluble form of the membrane-bound receptor uPAR, which is bound to a variety of immune cells, smooth muscle cells, and podocytes by a glycosyl-phosphatidylinositol (GPI) anchor[2,3]. uPAR is a receptor for urokinase-type plasminogen activator (uPA), an enzyme known for activating plasminogen into plasmin[4]. Plasmin's proteolysis of extracellular matrices (ECMs) is essential for fibrin blood clot degradation and clearance[4]. Other than participating in the plasminogen activator system, uPAR plays a role in various cellular processes including cell adhesion, migration, proliferation, angiogenesis, and chemotaxis[2,5]. uPAR can be cleaved into its soluble form, suPAR, by several proteases, including uPA, GPI-specific phospholipase D, matrix metalloproteinases (MMPs), cathepsin G, neutrophil elastase, and plasmin[6].

In the general population, an elevated plasma suPAR level has been found to predict various health conditions, including incident cancer, cardiovascular disease, diabetes, depression, as well as early mortality[7–9]. Elevated suPAR levels are also associated with pulmonary diseases including asthma and chronic obstructive pulmonary disease (COPD)[10–13]. It is also known that increased suPAR levels have strong associations with chronic kidney disease (CKD) across populations[14–17], and have been able to independently predict declining eGFR (estimated glomerular filtration rate) and incident CKD[18]. Mouse models have indicated that suPAR may not only be associated with acute kidney injury, but may be causative in the development of this[19]. In acute medically ill patients, increased suPAR is associated with readmissions and mortality, independent of clinical presentation, and suPAR is used in clinical routine in some European emergency departments for patient risk assessment[20].

Lifestyle factors associated with suPAR have been extensively studied. In particular, smoking has been found to be strongly associated with higher suPAR levels, and suPAR can be lowered by smoking cessation[21]. Unhealthy diet, inactive lifestyle, and obesity have substantial impacts on suPAR levels in the general population[9,22,23]. In addition, longitudinal research shows that multiple childhood risk factors (including exposure to adverse experiences, low IQ, and poor self-control) are associated with elevated suPAR in adulthood[24].

The genetics of suPAR remain unexplored but may shed light on the biology of the marker and its connection to outcomes. A recent genome-wide association analysis (GWAS) meta-analysis of the well-known inflammatory marker C-reactive protein (CRP) identified 58 associated genetic loci and consequently provided new insight into the genetic etiology of chronic inflammation[25]. However, CRP and suPAR reflect different aspects of chronic inflammation despite both being used as inflammatory biomarkers[26]. In addition, unlike CRP, suPAR is a stable biomarker as circadian changes in plasma suPAR are minimal[27–29], and suPAR measurements in individuals have been shown to be correlated across five and seven years[23,30]. We aimed to investigate whether suPAR plasma levels are under the genetic influence and if so, identify associated genetic variants that may facilitate our understanding of suPAR's biology and its links to associated diseases. Moreover, gaining new insight into suPAR through genetics may potentially improve the marker's current prognostic capabilities.

Therefore, we performed a heritability analysis in a sample of British twins to estimate the genetic contribution to suPAR levels

for the first time. We then performed a GWAS on suPAR levels in a general Icelandic population cohort and in a population of healthy Danish blood donors and combined these in a meta-analysis to identify genetic variants that affect this chronic inflammation marker's plasma levels. Significant findings were followed-up in two independent cohorts, a sample from Great Britain and another from New Zealand. Furthermore, a pathway-based analysis as well as phenome-wide association studies (pheWAS) were performed to examine the suPAR-associated variants and their predicted genes further.

## Results

**Heritability of suPAR.** We tested if variation in suPAR levels at age 18 years was genetically influenced in the Environmental Risk (E-Risk) Longitudinal Twin Study sample (Great Britain). Within-pair correlations for suPAR levels were $r = 0.69$ (95% CI: 0.65–0.73) for MZ twin pairs, and $r = 0.39$ (95% CI: 0.32–0.46) for DZ twin pairs. Using a univariate twin model, we found that additive genetic effects accounted for 60% (95% CI: 38–82%) of the variation in suPAR levels, while shared environmental influences accounted for 10% (95% CI: 0–31%) of the variance and nonshared environmental influences accounted for 30% (95% CI: 26–35%) of the variance in suPAR levels. We additionally calculated the SNP-based heritability based on the general Icelandic population cohort. The SNP heritability estimate was calculated to be 12.5% (SD: 4.8%).

**GWAS meta-analysis.** We performed GWASs on plasma suPAR levels in the general Icelandic population cohort ($N = 35,559$) as well as in the Danish Blood Donor Study (DBDS) ($N = 12,177$). We performed a meta-analysis of the two GWASs ($N = 47,736$) and employed a weighted Bonferroni adjustment to determine statistical significance as previously described[31]. The P-value significance thresholds were $2.0 \times 10^{-7}$ for high-impact variants (including stop-gained, frameshift, splice-acceptor, or splice-donor variants, $N = 11,723$), $4.0 \times 10^{-8}$ for 'moderate-impact' variants (including missense, splice-region variants, and in-frame indels, $N = 202,336$), $3.7 \times 10^{-9}$ for 'low-impact' variants (including upstream and downstream variants, $N = 2,896,354$), and $6.1 \times 10^{-10}$ for the 'lowest-impact' variants (including intron and intergenic variants, $N = 37,239,641$). Our GWAS meta-analysis identified 13 independent genome-wide significant genetic variants associated with suPAR across 11 distinct loci in the genome (Fig. 1, Table 1, Supplementary Data 1). The variants were tested for heterogeneity between the two cohorts. Only one variant, the rs71311394 intron variant in *ST3GAL6*, shows evidence of heterogeneity at $P < 0.05$. However, the direction of effects for rs71311394 are consistent between the two cohorts and the association with suPAR levels is significant in each cohort (Effect$_{Ice} = 0.11$, $P_{Ice} = 5.3 \times 10^{-12}$; Effect$_{DK} = 0.21$, $P_{DK} = 2.1 \times 10^{-18}$). Two of the 13 genetic variants (rs114821641 and rs755902185 located in the *PLA2R1/LY75* locus) were identified via conditional analysis using the Icelandic data exclusively, where linkage disequilibrium (LD) data is available for the same population.

**Comparison of genetic variants' effects for suPAR unadjusted vs adjusted for smoking.** As suPAR levels have strong associations with smoking, we investigated whether smoking status would affect the outcome of the suPAR GWAS results. Using the Icelandic cohort, we performed two GWASs; the 30,469 individuals with available information on smoking status, unadjusted for smoking; and the same 30,469 individuals, adjusted for

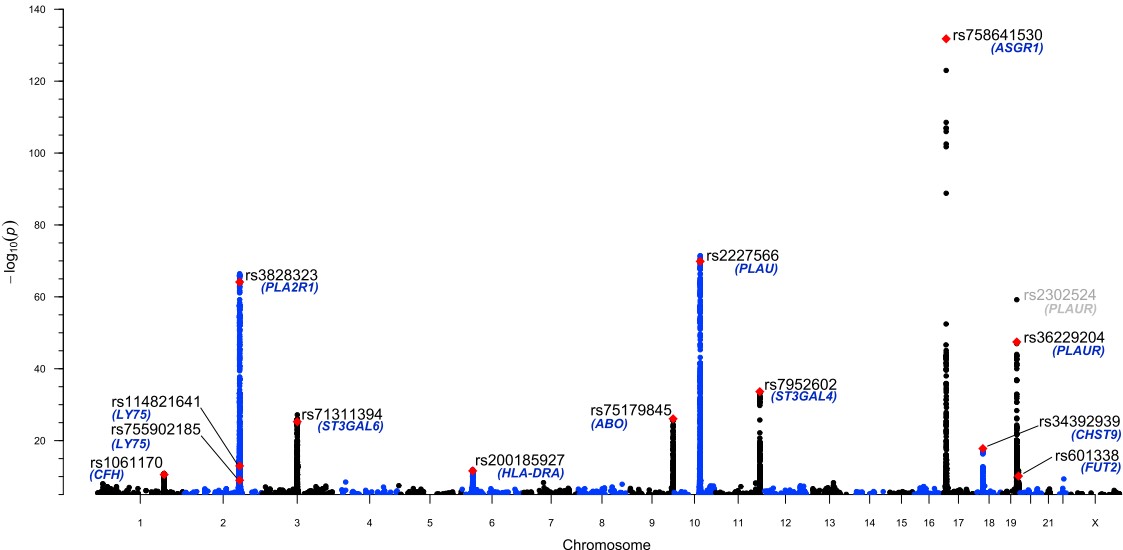

**Fig. 1 suPAR GWAS meta-analysis Manhattan plot (N = 47,736), showing the 11 genome-wide significant loci and the 13 independently significant variants associated with suPAR.** The negative log10 transformed *P* values for variants are plotted by chromosomal location. *Y* axis begins at $P = 1 \times 10^{-5}$. Red points signify the 13 independently significant variants associated with suPAR, with variant IDs annotated in black and the corresponding gene annotated in blue.

smoking. Smokers had higher suPAR levels than non-smokers (Effect = 0.12 SD, $P = 2.2 \times 10^{-23}$ from *t*-test). A test of difference in the GWAS results between the two above-mentioned Icelandic GWASs revealed no difference when adjusting for smoking, with heterogeneity *p*-values ranging from 0.94 to 1.00 (Supplementary Data 2).

**Validation of variants from independent cohorts**. The lead genetic variants for each suPAR-associated locus were examined in two independent validation cohorts (Table 2 and Supplementary Data 3). We used a sample consisting of 837 individuals of white European-descent non-Maori descent from The Dunedin Multidisciplinary Health and Development Study (Dunedin) cohort, of which eight of the 13 variants were available for replication. A sample of 1444 E-Risk members of white European-descent was also used as a validation cohort, of which six of the 13 variants were available. *P*-values < 0.05 were considered statistically significant in the validation phase. In the Dunedin cohort, five out of the eight available variants were confirmed and all eight variants had effect estimates in the same direction. In the E-Risk cohort, three out of the six available variants were confirmed and five variants had effect estimates in the same direction.

**Summary of suPAR-associated loci**. The following section lists the eleven loci in further detail, describing the variants' predicted genes based on their position within a given gene or the closest gene, associations from previous studies, and selected annotated gene ontology (GO) biological processes for each gene.

**Chr1.q31.3**. The missense variant rs1061170-C on chromosome 1 in the gene *CFH* is associated with an increase in suPAR (effect estimate = 0.048 units of a standard deviation per copy increment in the effect allele; $P = 2.79 \times 10^{-11}$, effect allele frequency (EAF) = 0.39). The variant causes a missense mutation (His > Tyr) in the gene *CFH*, which encodes the glycoprotein Complement Factor H; a protein that regulates complement activation in an immune response. The variant is known to be associated with

age-related macular degeneration[32,33]. GO terms biological processes associated with the gene include complement activation, regulation of complement-dependent cytotoxicity, and viral process.

**Chr2.q24.2**. Three independently significant variants at a locus on chromosome 2 were found, resulting in two candidate genes in this locus. The missense variant rs3828323-C in the gene *PLA2R1* (phospholipase A2 receptor 1) is associated with an increase in suPAR (effect = 0.118; $P = 7.5 \times 10^{-65}$, EAF = 0.48). Several variants in *PLA2R1* have previously been associated to membranous nephropathy, and serum anti-PLA2R1 antibody associates with loss of kidney function[34,35]. GO terms biological processes associated with the gene include cytokine production, negative regulation of phospholipase A2 activity, and receptor-mediated endocytosis among others.

The remaining two significant variants in this locus are less common and are located in the nearby gene *LY75* encoding the protein lymphocyte antigen 75. rs114821641-T causes a stop-gain mutation and an increase in suPAR (effect = 0.400; $P = 1.08 \times 10^{-13}$, EAF = 0.003), and rs755902185 is a deletion which causes a frameshift mutation in *LY75* and an increase in suPAR (effect = 0.445; $P = 1.05 \times 10^{-9}$, EAF = 0.0004). Variants in LY75 have previously been associated with Inflammatory Bowel Syndrome and Crohn's disease[36,37]. GO terms biological processes associated with the gene include endocytosis, immune response, and inflammatory response.

**Chr3.q12.1**. The 3 prime untranslated region (UTR) variant rs71311394-G on chromosome 3 in the gene *ST3GAL6* (ST3 Beta-Galactoside Alpha-2,3-Sialyltransferase 6) is associated with an increase in suPAR (effect = 0.137; $P = 5.31 \times 10^{-26}$). GO terms biological processes associated with the gene include glycolipid biosynthetic processes, cellular response to interleukin-6, and glycosylation among others[38].

**Chr6.p21.32**. The deletion variant rs200185927 on chromosome 6 downstream from *HLA-DRA* is associated with an increase in

**Table 1 Summary statistics for the 13 independently genome-wide significant variants from the meta-analysis (N = 47,736).**

| Variant ID | Chr | Position (Hg38) | Effect allele | Other allele | Effect-allele frequency | Gene | Variant type | Combined effect | 95% CI | Comb. P-value |
|---|---|---|---|---|---|---|---|---|---|---|
| rs1061170 | 1 | 196690107 | C | T | 0.39 | CFH | Missense | 0.048 | 0.034, 0.062 | $2.79 \times 10^{-11}$ |
| rs3828323 | 2 | 159951564 | C | T | 0.48 | PLA2R1 | Missense | 0.118 | 0.104, 0.132 | $7.50 \times 10^{-65}$ |
| rs114821641 | 2 | 159858447 | T | C | 0.003 | LY75 | Stop gained | 0.400 | 0.294, 0.506 | $1.08 \times 10^{-13}$ |
| rs755902185 | 2 | 159864896 | A | AC | 0.0004 | LY75 | Frameshift | 0.445 | 0.302, 0.588 | $1.05 \times 10^{-9}$ |
| rs71311394 | 3 | 98793766 | G | A | 0.06 | ST3GAL6 | 3 Prime UTR | 0.137 | 0.112, 0.162 | $5.31 \times 10^{-26}$ |
| rs200185927 | 6 | 32449458 | A | AAAGAAGAAAG | 0.25 | HLA-DRA | Downstream | 0.060 | 0.043, 0.077 | $2.58 \times 10^{-12}$ |
| rs75179845 | 9 | 133257567 | C | T | 0.10 | ABO | Intron | 0.148 | 0.121, 0.175 | $8.37 \times 10^{-27}$ |
| rs2227566 | 10 | 73913973 | C | T | 0.46 | PLAU | Splice region | −0.124 | −0.138, −0.110 | $1.36 \times 10^{-70}$ |
| rs7952602 | 11 | 126363774 | C | G | 0.14 | ST3GAL4 | Intron | −0.131 | −0.152, −0.110 | $2.33 \times 10^{-34}$ |
| rs758641530 | 17 | 7176936 | C | CCCCCAGCCCCAG | 0.004 | ASGR1 | Intron | 1.089 | 1.002, 1.176 | $1.66 \times 10^{-132}$ |
| rs34392939 | 18 | 27113190 | GAAA | GAA | 0.30 | CHST9 | Intron | −0.065 | −0.080, −0.050 | $1.58 \times 10^{-18}$ |
| rs36229204 | 19 | 43671830 | T | C | 0.03 | PLAUR | Upstream | −0.270 | −0.306, −0.234 | $3.55 \times 10^{-48}$ |
| rs601338 | 19 | 48703417 | G | A | 0.55 | FUT2 | Stop gained | −0.046 | −0.060, −0.032 | $7.21 \times 10^{-11}$ |

Effect estimate in units of standard deviation per copy increment in the effect allele.
Allele frequencies reported are based on European populations (Supplementary Data 1 reports the cohorts-specific effect allele frequencies).
Table is ordered by genomic locus. Full table is available in Supplementary Data 1.

suPAR (effect = 0.06; $P = 2.58 \times 10^{-12}$, EAF = 0.25). GO terms biological processes associated with this gene include antigen processing and presentation of exogenous peptide antigen via major histocompatibility complex class II among other immune response processes.

**Chr9.q34.2.** Intron variant rs75179845-C on chromosome 9 in the gene *ABO* is associated with an increase in suPAR (effect = 0.148; $P = 8.37 \times 10^{-27}$, EAF = 0.10). The gene determines the blood group of the individual, and GO terms biological processes associated with the gene include lipid and protein glycosylation[38].

**Chr10.q22.2.** Splice-site variant rs2227566-C on chromosome 10 in the gene *PLAU* is associated with a decrease in suPAR (effect = −0.124; $P = 1.36 \times 10^{-70}$, EAF = 0.46). The variant is located in the splice region of *PLAU*, the gene encoding the protein uPA, which can bind to uPAR and consequently release the receptor into the bloodstream as suPAR. The protein uPA cleaves plasminogen to form the active form of plasmin and GO terms biological processes associated with this gene include blood coagulation, chemotaxis, fibrinolysis, regulation of smooth muscle cell-matrix adhesion, and migration among others. The lead variant rs2227566-C has been previously associated with asthma and airway hyperresponsiveness[39].

**Chr11.q24.2.** Intron variant rs7952602-C on chromosome 11 in the gene *ST3GAL4* (ST3 Beta-Galactoside Alpha-2,3-Sialyltransferase 4) is associated with a decrease in suPAR (effect = −0.131; $P = 2.33 \times 10^{-34}$, EAF = 0.14). GO terms biological processes of this gene include glycolipid biosynthetic processes and glycosylation among others[38].

**Chr17.p13.1.** Deletion variant rs758641530 on chromosome 17 in the gene *ASGR1* (asialoglycoprotein receptor) is associated with an increase in suPAR (effect = 1.089; $P = 1.66 \times 10^{-132}$, EAF = 0.004). ASGR1 is a lectin that mediates the endocytosis of plasma glycoproteins, and a previous study found that a deletion in this gene is associated with reduced levels of non-HDL cholesterol and a reduced risk of coronary artery disease[40].

**Chr18.q11.2.** Intron variant rs34392939 on chromosome 18 in the gene *CHST9* (Carbohydrate Sulfotransferase 9) is associated with a decrease in suPAR (effect = −0.065; $P = 1.58 \times 10^{-18}$, EAF = 0.30). A GO term biological process associated with this gene is proteoglycan biosynthetic process[38], and copy number variations of *CHST9* associate with hematologic malignancies[41].

**Chr19.q13.31.** Variant rs36229204-T on chromosome 19 in the upstream from *PLAUR* (encodes uPAR) is associated with a decrease in suPAR (effect = −0.270; $P = 3.55 \times 10^{-48}$, EAF = 0.03). The variant is in high LD with rs4251805 ($R^2 = 1$; $D' = 1$) which exists in *PLAUR's* 5′ UTR. GO terms biological processes associated with the gene include blood coagulation, chemotaxis, fibrinolysis, and regulation of proteolysis among others.

In addition, we identified *PLAUR* missense variant rs2302524-C to be independently significant in both the Danish and Icelandic GWASs, but with opposite effect directions (Effect$_{ICE}$ = −0.24; $P_{ICE} = 3.29 \times 10^{-113}$, EAF$_{ICE}$ = 0.18; Effect$_{DK}$ = 0.09, $P_{DK} = 4.68 \times 10^{-8}$, EAF$_{DK}$ = 0.17). Due to the inconsistent direction, it has not been included as a valid 14th suPAR-associated signal and hence is not shown in Table 1. Validation cohorts from New Zealand and Great Britain indicate that rs2302524 is significantly associated with increased suPAR levels as found in the Danish cohort (Supplementary Data 4). The

**Table 2 Results from the validation cohorts of the suPAR-associated variants.**

**suPAR GWAS Meta-Analysis Summary Table**

| Variant ID | Chr | Position (Hg38) | EA | EAF | Gene | Comb. effect | Comb. P-value | Dunedin (New Zealand) | | | | | E-Risk (Great Britain) | | | | |
|---|---|---|---|---|---|---|---|---|---|---|---|---|---|---|---|---|---|
| | | | | | | | | N | EA | EAF | Effect | p value | N | EA | EAF | Effect | p value |
| rs1061170 | 1 | 196690107 | C | 0.39 | CFH | 0.048 | 2.79E–11 | 829 | C | 0.37 | **0.092** | **0.050** | 1315 | C | 0.48 | **0.112** | **0.004** |
| rs3828323 | 2 | 159951564 | C | 0.48 | PLA2R1 | 0.118 | 7.50E–65 | 829 | C | 0.49 | 0.032 | 0.481 | | | | | |
| rs114821641 | 2 | 159858447 | T | 0.003 | LY75 | 0.400 | 1.08E–13 | | | | | | | | | | |
| rs755902185 | 2 | 159864896 | A | 0.0004 | LY75 | 0.445 | 1.05E–09 | | | | | | | | | | |
| rs71311394 | 3 | 98793766 | G | 0.06 | ST3GAL6 | 0.137 | 5.31E–26 | 827 | G | 0.08 | **0.253** | **0.003** | 1308 | G | 0.07 | −0.134 | 0.038 |
| rs200185927 | 6 | 32449458 | A | 0.25 | HLA-DRA | 0.060 | 2.58E–12 | | | | | | | | | | |
| rs75179845 | 9 | 133257567 | C | 0.1 | ABO | 0.148 | 8.37E–27 | 829 | C | 0.07 | **0.178** | **0.039** | 1413 | C | 0.07 | **0.083** | 0.305 |
| rs2227566 | 10 | 73913973 | C | 0.46 | PLAU | −0.124 | 1.36E–70 | 829 | C | 0.46 | **−0.155** | **0.001** | | | | | |
| rs7952602 | 11 | 126363774 | C | 0.14 | ST3GAL4 | −0.131 | 2.33E–34 | 796 | C | 0.13 | −0.018 | 0.800 | 1273 | C | 0.12 | **−0.041** | 0.507 |
| rs754165241 | 17 | 7176936 | GAAA | 0.004 | ASGR1 | 1.089 | 1.66E–132 | | | | | | | | | | |
| rs34392939 | 18 | 27113190 | C | 0.3 | CHST9 | −0.065 | 1.58E–18 | | | | | | | | | | |
| rs36229204 | 19 | 43671830 | T | 0.03 | PLAUR | −0.270 | 3.55E–48 | 829 | T | 0.02 | **−0.299** | **0.035** | 1415 | T | 0.03 | **−0.248** | **0.008** |
| rs601338 | 19 | 48703417 | G | 0.55 | FUT2 | −0.046 | 7.21E–11 | 829 | G | 0.49 | −0.001 | 0.978 | 1417 | G | 0.52 | **−0.086** | **0.026** |

Results in bold signify effect estimates in the same direction and/or significant P-values.
N number of samples, EA effect allele, EAF effect allele frequency.

variant has previously been associated with worse baseline lung function (FEV1) in smokers as well as an increased risk of asthma and worse FEV1 in individuals with asthma[42,43].

**Chr19.q13.33.** Stop-gain mutation variant rs601338-G on chromosome 19 in the gene *FUT2* (Fucosyltransferase 2) is associated with a decrease in suPAR (effect = −0.046; $P = 7.21 \times 10^{-11}$, $EAF_{ICE} = 0.39$; $EAF_{DK} = 0.55$). GO terms biological processes associated with the gene include protein glycosylation, L-fucose catabolic process, and regulation of cell adhesion among others.

In summary, we identified 13 genome-wide significant suPAR-associated variants and based on literature searches and GO term annotations, the variants are found in and around 12 genes encoding uPAR/suPAR (*PLAUR*) and its ligand uPA (*PLAU*), genes with relations to glycoprotein biosynthesis and glycosylation (*ASGR1, ST3GAL4, ST3GAL6, ABO, CHST9, FUT2*), genes involved in immune response (*LY75, HLA-DRA, CFH*), and *PLA2R1* (of which variants have been previously associated with membranous nephropathy).

**Overrepresentation of biological processes in the suPAR-GWAS-associated set of genes.** We used the Biological Network Gene Ontology (BiNGO) bioinformatics tool[44] to quantitatively assess whether there are GO terms (biological processes) that are statistically overrepresented in our set of 12 suPAR-associated genes. Analysis of the 12 suPAR-associated genes using the BiNGO tool revealed 39 GO term biological processes significantly overrepresented after multiple test corrections. Due to the hierarchical nature of the GO term gene sets, the majority of the 39 biological processes are overlapping and can therefore be grouped into approximately nine biological process branches (Fig. 2). Significant biological processes that are overrepresented include glycoprotein biosynthetic process ($P = 8.85 \times 10^{-8}$), protein amino acid glycosylation ($P = 2.14 \times 10^{-6}$), skeletal muscle tissue regeneration ($P = 3.53 \times 10^{-5}$), endocytosis ($P = 7.31 \times 10^{-4}$), response to wounding ($P = 7.86 \times 10^{-4}$), fibrinolysis ($P = 5.86 \times 10^{-3}$), attachment of GPI anchor to protein ($P = 5.86 \times 10^{-3}$), L-fucose catabolic process ($P = 7.52 \times 10^{-3}$) and chemotaxis ($P = 8.47 \times 10^{-3}$) (Supplementary Data 5).

**suPAR polygenic risk scores (PRSs) and pheWASs.** To investigate whether the combined effect of suPAR-associated genetic variations were associated with specific phenotypes, we performed pheWASs using suPAR PRSs as the exposure. We calculated PRSs for the individuals in the Icelandic population cohort based on the summary statistics from the Danish cohort's suPAR GWAS. The PRSs explained 0.94% of the suPAR variance in Icelandic individuals.

A total of 14,493 case/control phenotypes and 28,389 quantitative phenotypes in the Icelandic population cohort were tested. After Bonferroni multiple testing correction ($P < 7.86 \times 10^{-7}$), in the case-control pheWAS, we found that suPAR PRSs were associated with type 1 diabetes (effect = 2.212; $P = 5.11 \times 10^{-18}$), autoimmune diseases (effect = 0.354; $P = 4.30 \times 10^{-10}$) and obesity (effect = 0.634; $P = 1.04 \times 10^{-7}$). In the quantitative phenotype pheWAS, we found that suPAR PRSs were associated with increased levels of plasma PLA2R1 (effect = 0.649; $P = 1.99 \times 10^{-105}$), increased levels of B12 (effect = 0.125; $P = 7.30 \times 10^{-13}$), decreased high-density lipoprotein cholesterol (effect = −0.107; $P = 3.96 \times 10^{-10}$), increased fasting plasma glucose (effect = 0.089; $P = 6.04 \times 10^{-9}$), increased alkaline phosphatase (effect = 0.100; $P = 1.82 \times 10^{-8}$), increased potassium (effect = 0.085; $P = 2.61 \times 10^{-9}$), increased BMI (effect = 0.109; $P = 1.03 \times 10^{-7}$) and increased eosinophils (effect = 0.056; $P = 7.01 \times 10^{-8}$) (Supplementary Data 6–7).

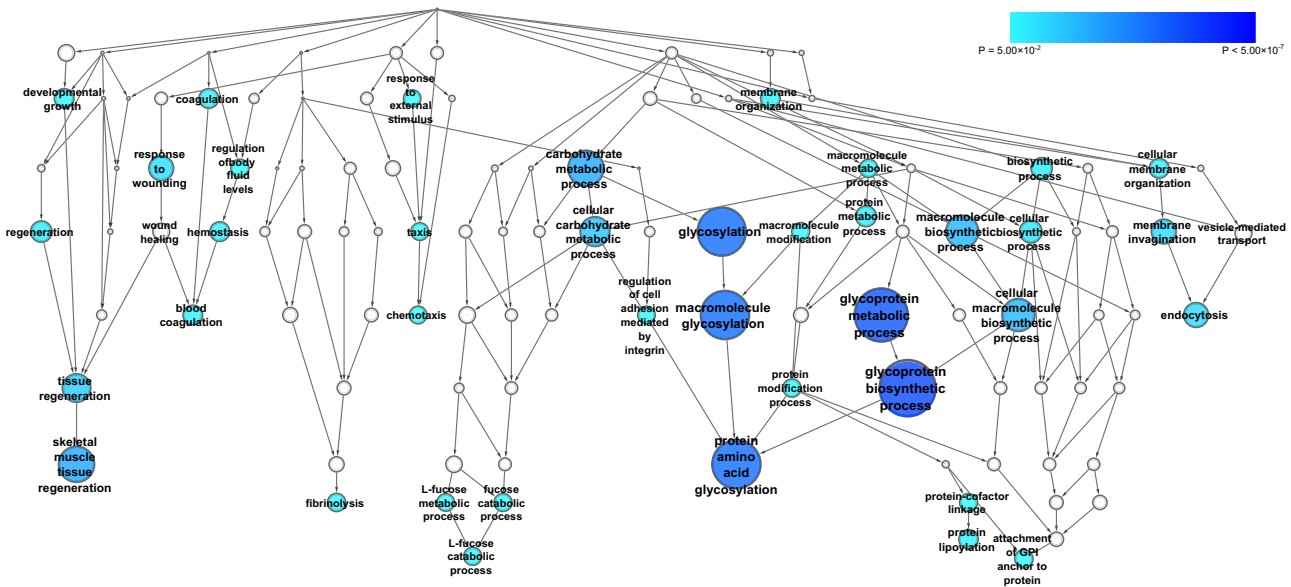

**Fig. 2 Results of the BiNGO pathway over-representation analysis for GO biological processes, using the 12 genes found from the suPAR GWAS meta-analysis.** Significant overrepresented pathway gene sets ($P < 0.05$ after surviving Benjamini & Hochberg False Discovery Rate correction) are shaded blue with size and darker shades signifying lower $P$-values as depicted in the legend.

We additionally calculated PRSs for the individuals in the UK Biobank based on the summary statistics from the suPAR GWAS meta-analysis (Icelandic cohort + Danish cohort). 15,120 case/control phenotypes and 5,609 quantitative phenotypes were available in the UK Biobank pheWAS. No significant case/control phenotypes associated with suPAR after Bonferroni multiple test correction, but for the quantitative phenotypes, increased mean corpuscular hemoglobin (effect = 0.006; $P = 2.40 \times 10^{-11}$) and increased mean corpuscular volume (effect = 0.005; $P = 3.80 \times 10^{-9}$) were significant (Supplementary Data 8). Single-variant pheWASs were also performed for each of the 13 independently significant variants separately in both the Icelandic and UK Biobank dataset using the same methods and the significant results are available in Supplementary Data 9–12.

**Mendelian randomization analyses for suPAR vs. the identified pheWAS findings.** We performed Mendelian randomization analyses for suPAR and the identified pheWAS findings using phenotypes available in the Icelandic population cohort. No significant results were found and the removal of outliers detected using MR-PRESSO did not improve the estimates significantly (Supplementary Data 13).

**Cis-expression quantitative trait loci (eQTL) analysis of the suPAR-associated variants.** We tested if our variants were in high LD ($r^2 > 0.8$) with one more top cis-eQTL based on various tissues and 17 sources including GTEx and Icelandic data. Results are reported in Supplementary Data 14 and sources listed in Supplementary Data 15.

**Genetic Correlation between suPAR and CRP.** We performed a genetic correlation analysis between suPAR and CRP. The genetic correlation between suPAR and CRP was calculated to be 0.2351 (SD = 0.0533, $P = 1.03 \times 10^{-5}$), using suPAR measurements from the Icelandic/Danish meta-analysis and CRP measurements from UK Biobank. We have additionally checked for CRP variants from the GWAS-catalog for our suPAR-associated variants and their LD-classes (all variants with $r^2 > 0.80$). Only the *FUT2*

variant (rs601338) was found to be an overlapping variant, where the same variant was reported in Han et al.[45]. We have additionally searched all the genes that we report to see if they have been reported with CRP, of which the only overlap we find in addition to *FUT2* is *ABO*, where there is a variant at chr9:133266942 (rs643434) reported to associate with CRP in the CRP GWAS meta-analysis by Ligthart et al.[25]. This variant has $r^2 = 0.17$ with the *ABO* variant found in our study.

### Discussion
Eleven genome-wide significant loci driven by 13 variants were associated with suPAR in our GWAS meta-analysis based on 47,736 individuals. These findings, including our heritability estimate of 60%, provide an updated explanation for the inter-individual variation in suPAR plasma levels.

We identified suPAR-associated variants localized in and near the genes encoding uPAR (*PLAUR*) and the ligand uPA (*PLAU*), although how they affect suPAR plasma levels is uncertain. The lead variant rs36229204 at the *PLAUR* locus is in complete LD with rs4251805 ($r^2 = 1$; $D' = 1$) which is located in *PLAUR*'s 5' UTR, a region known for its importance in the regulation of a gene's translation. Similarly, the lead variant rs2227566 at the *PLAU* locus is located in the splice region of the gene *PLAU*, but it is also in LD ($r^2 = 0.44$; $D' = 1$) with missense variant rs2227564-T (meta-analysis effect = −0.131; $P = 1.57 \times 10^{-62}$, EAF = 0.26) which is the lead variant in the Danish cohort's suPAR GWAS. This *PLAU* variant has previously been associated with an increased risk of colorectal cancer, asthma, oral tongue squamous cell carcinoma, and poor coronary collateral circulation in coronary artery disease patients, as well as Alzheimer's Disease[39,46–49]. The variant causes a missense mutation in the kringle domain of uPA—the important domain necessary for protein–protein interactions including integrins[50]. The binding of uPA and suPAR is stabilized by the kringle domain[51,52], suggesting that this missense mutation may produce a conformational change that reduces uPA's ability to bind and/or to cleave uPAR, thereby reducing suPAR levels as seen in the results from this study.

Our results may also provide some insight into suPAR's possible role in chronic and acute kidney disease[18,19]. Significant variants were found in and near *PLA2R1*, where the top variant rs3828323 increases suPAR levels. As a missense variant, the resulting change from non-polar to polar residue may alter the binding of *PLA2R1's* ligand PLA2 (secretory phospholipase A2). The residue change is located in the extracellular part of the receptor between two C-type lectin domains, of which one is part of the receptor-binding region for its ligand PLA2. Interestingly, several variants in *PLA2R1* have previously been associated to membranous nephropathy, and serum anti-PLA2R1 antibody associates with loss of kidney function[34,35]. Two studies on membranous nephropathy have identified associated genetic variants, of which one found an association with two variants: rs3828323 and rs35771982[34], and the other study identified had rs17831251 as the lead associated variant[53]. The two variants that are not reported in our study, i.e., rs35771982 and rs17831251 associate with suPAR in our meta-analysis but these associations do not hold up after adjusting for our lead variant ($r^2 = 0.32$ and $r^2 = 0.19$ respectively). This would indicate that the signal reported in these previous membranous nephropathy genetic studies at the *PLA2R1* locus is the same as for suPAR in our study. It is well-reported that increased suPAR has strong associations with CKD[14–17], though it is unknown how suPAR and PLA2R1 are related to each other in respect to kidney disease. It is possible that when anti-PLA2R1 attaches to podocyte-bound PLA2R1 due to the altered structure of the receptor caused by the genetic variant, it forms immune complexes which consequently activates the immune system. As a result, the inflammatory response would increase suPAR levels and allow suPAR to activate beta3 integrins on the podocytes as shown in the previous studies[3,54,55], and this activation would produce the podocyte conformational change that causes membranous nephropathy. It may therefore be hypothesized that it is a variant in *PLA2R1* that gives suPAR the appearance of a causal role in the development of membranous nephropathy and other kidney diseases previously shown to have strong associations with suPAR levels. Though rs3828323 in *PLA2R1* was the lead variant at this locus in our meta-analysis, it should be noted that two additional independent but rare variants were identified in this locus, located in the gene *LY75*. The *PLA2R1* variant rs3828323 exists in an LD-class which overlaps into both genes. *LY75* and *PLA2R1* may therefore both be considered candidate genes in this locus. However, from the PRS-based quantitative pheWAS, we found that a higher suPAR PRS was strongly associated with increased PLA2R1 plasma levels (effect = 0.649; $P = 1.99 \times 10^{-105}$), further supporting *PLA2R1* as a gene of interest for future studies.

Another noteworthy observation from our study, supported by the results from the BiNGO pathway-based analysis, is that six of the suPAR-associated loci have significant variants in genes encoding proteins that are in some form involved in processes related to glycoprotein biosynthesis or protein glycosylation. These genes include *ST3GAL4*, *ST3GAL6*, *ABO*, *CHST9*, *ASGR1*, and *FUT2*. From the Kyoto Encyclopedia of Genes and Genomes (KEGG) database[56] it is also evident that the proteins ST3GAL4, ST3GAL6, ABO and FUT2 participate in the glycosphingolipid biosynthesis (lacto and neolacto series) pathway. Glycosphingolipids are often localized in glycosphingolipid-enriched microdomains called lipid rafts where they have a role in mediating cell–cell interactions and regulating proteins in the same plasma membrane[57]. Glycosphingolipids may therefore have a regulatory function on suPAR as it has previously been found that uPA-induced uPAR cleavage is strongly accelerated when uPAR is localized in lipid rafts[58]. ASGR1 is a lectin that mediates the endocytosis of plasma glycoproteins which may also impact uPAR's role on the plasma membrane. It is well known that

uPAR is highly glycosylated[59,60] but the function of the glycosylation is not completely understood. However, glycosylation has been found to increase uPAR's affinity to uPA and enhance suPAR's solubility[60–62]. Though the results from our study suggest that uPAR glycosylation may have a molecular function that impacts suPAR plasma levels, the effects of glycosylation and deglycosylation on suPAR detection for the two methods used in this study are not known, and this may affect the results. Nevertheless, the results support previous studies that glycosylation increases uPAR-uPA affinity and suPAR solubility and therefore the genes identified in our study may be considered in future candidate gene studies to investigate their role in affecting the amount of suPAR present in plasma.

Two different suPAR detection methods were used in our study; a proteomics-based assay in the Icelandic population and ELISA in the Danish population. Although a direct comparison between the two different assays using the same samples was not possible, a comparison of the results of the GWAS findings between the Danish and Icelandic population cohorts was performed, of which the results showed high concordance. Only one variant, the rs71311394 variant in *ST3GAL6*, shows evidence of heterogeneity ($P = 0.003$). However, the direction of effects for rs71311394 are consistent between the two populations (Effect$_{Ice}$ = 0.11 vs. Effect$_{DK}$ = 0.21) and the association with suPAR levels is genome-wide significant in both populations ($P_{Ice} = 5.3 \times 10^{-12}$ vs. $P_{DK} = 2.1 \times 10^{-18}$). Given this high degree of similarity in the GWAS findings we believe that a direct comparison between the assays would not add any further genetic insights into our findings.

As our findings indicate that individuals can be genetically predisposed to higher or lower suPAR levels, it may have implications for future precision or personalized medicine practices by potentially improving suPAR's current prognostic capabilities. It is known that increased suPAR is associated with readmissions and mortality in acute medically ill patients, and suPAR is used in patient risk assessments in some Danish hospitals[20]. Genetic profiling of patients may contribute to optimized patient treatment by identifying patients genetically predisposed to higher or lower levels of chronic inflammation, thereby enabling risk assessment of a patient at an earlier stage before they reach an acute medically ill stage of their disease progress. However, it must be noted that the SNP-based heritability based on the Icelandic cohort was calculated to be 12.5% (SD: 4.9%). As our univariate twin model estimated that additive genetic effects account for 60% of the variation in suPAR levels, we are unable to account for much of the heritability of suPAR, a situation frequently reported in GWASs of other phenotypes[63]. Larger studies will likely enable the discovery of more variants that explain some of this missing heritability, as well as future studies focusing on other forms of genetic variation such as copy number variants.

In conclusion, we provide evidence that suPAR plasma levels are under the substantial genetic influence and that 13 independently significant genetic variants at 11 distinct loci influence suPAR levels in Icelandic and Danish individuals. Our data further support genetic links between suPAR-measured chronic inflammation and phenotypes such as diabetes and obesity. Our results indicate that suPAR's strong associations with chronic kidney disease may be related to a suPAR-associated missense variant in the gene *PLA2R1*, and that variants in many genes related to glycosylation and glycoprotein biosynthesis pathways affect suPAR levels. Genes identified in this study may be examined as candidate genes in future functional studies to help clarify suPAR's potential role in the causation of associated diseases, as well as the underlying mechanisms that give suPAR its prognostic value as a unique marker of chronic inflammation.

## Methods

**Participants**. The meta-analysis is based on data from two Northern European population cohorts: a Danish cohort consisting of healthy blood donors, and a general Icelandic population cohort.

*Danish cohort*. The Danish cohort is based on participants originating from the DBDS, a nationwide research platform utilizing the existing infrastructure in the Danish blood banks[64]. Participants must be generally healthy and not on medication to be eligible as donors. Upon enrollment, participants gave informed consent, whole blood, plasma, and answered a comprehensive questionnaire. So far, ~110,000 adult DBDS participants have been enrolled with informed consent, whole blood, plasma samples, questionnaire data, and genome-wide genotype data gathered from each[65]. suPAR was measured in 14,367 consecutive DBDS participants from 1 March 2010 until 10 December 2010, of which 12,177 (84.8%) participated in the GWAS after fulfilling quality control requirements. The project is approved by the Research Ethics Committees by the following three protocols: The DBDS (M-20090237), Genetics of healthy ageing (CVK-1700407), Family study on the genetics of healthy ageing (NVK-1803847). The project is approved by the Danish Data Protection Agency under the combined approval for health care research at The Capital Region of Denmark (P-2019-99).

*Icelandic cohort*. Plasma samples from 40,004 Icelanders were collected during 2000–2019. Fifty-two percent of the samples were collected as part of the Icelandic Cancer Project (ICP), while the remaining samples (48%) were collected as part of various genetic programs at deCODE genetics, Reykjavík, Iceland. In the ICP, all prevalent and newly diagnosed Icelandic cancer cases and their relatives were invited to participate in a comprehensive study of cancer, along with a control population, randomly selected from the National Registry. The median collection date for samples collected in conjunction with ICP was 1 July 2002, whereas the median collection date for other samples was 15 May 2015. All samples were measured using the SOMAscan platform (SomaLogic), containing 5284 aptamers providing a measurement of relative binding of the plasma sample to each of the aptamers in relative fluorescence units (RFU), corresponding to 4792 proteins, of which suPAR is included. After quality control, unique measurements for $N = 35,559$ individuals (88.9%) were used for GWAS. All participants who donated samples gave informed consent and the National Bioethics Committee of Iceland approved the study (VSN-15-198) which was conducted in agreement with conditions issued by the Data Protection Authority of Iceland. Personal identities of the participants' data and biological samples were encrypted by a third-party system (Identity Protection System), approved, and monitored by the Data Protection Authority.

Two independent cohorts agreed to validate the findings from this study: the Environmental Risk Longitudinal (E-Risk) Twin Study from Great Britain, as well as The Dunedin Longitudinal Study from New Zealand, of which the former was additionally used for the twin/heritability analysis.

*Environmental risk (E-Risk) longitudinal twin study*. Participants were members of the E-Risk longitudinal twin study, which tracks the development of a 1994-95 birth cohort of 2,232 British children[66]. Briefly, the E-Risk sample was constructed in 1999–2000, when 1116 families (93% of those eligible) with same-sex 5-year-old twins participated in home-visit assessments. This sample comprised 56% monozygotic (MZ) and 44% dizygotic (DZ) twin pairs; sex was evenly distributed within zygosity (49% male). The sample represents socioeconomic conditions in Great Britain, as reflected in the families' distribution on a neighborhood-level socioeconomic index (ACORN [A Classification of Residential Neighborhoods], developed by CACI Inc. for commercial use): 25.6% of E-Risk families live in "wealthy achiever" neighborhoods compared to 25.3% nationwide; 5.3% vs. 11.6% live in "urban prosperity" neighborhoods; 29.6% vs. 26.9% in "comfortably off" neighborhoods; 13.4% vs. 13.9% in "moderate means" neighborhoods; and 26.1% vs. 20.7% in "hard-pressed" neighborhoods. (E-Risk underrepresents "urban prosperity" neighborhoods because such households are often childless.) Home visits were conducted when participants were aged 5, 7, 10, 12, and most recently, 18 years (93% participation). At age 18, each twin was interviewed by a different interviewer. Whole blood was collected from 82% ($n = 1700$) of the participants. Plasma was available for 1448 participants. The Joint South London and Maudsley and the Institute of Psychiatry Research Ethics Committee approved each phase of the study. Parents gave informed consent and twins gave assent between 5 and 12 years and then informed consent at age 18.

*The Dunedin multidisciplinary health and development study*. Participants were members of the Dunedin study, a longitudinal investigation of health and behavior in a representative birth cohort. Participants ($n = 1037$; 91% of eligible births; 52% male) were all individuals born between April 1972 and March 1973 in Dunedin, New Zealand (NZ), who were eligible based on residence in the province and who participated in the first assessment at age 3 years[67]. The cohort represented the full range of socioeconomic status (SES) in the general population of NZ's South Island and as adults matched the NZ National Health and Nutrition Survey on key adult health indicators (e.g., body mass index, smoking, GP visits) and the NZ Census of citizens of the same age on educational attainment. The cohort is primarily white (93%), matching South Island demographics[67]. Assessments were carried out at

birth and ages 3, 5, 7, 9, 11, 13, 15, 18, 21, 26, 32, and 38 years. At age 38 years, 95% ($n = 961$) of the 1007 participants still alive took part. At each assessment, each participant was brought to the research unit for interviews and examinations. Blood from participants of Maori ancestry was not transported to Duke University for cultural reasons, and plasma samples were not available for participants who did not provide blood or due to phlebotomy or defrost cycle problems. The relevant ethics committees approved each phase of the Study and written informed consent was obtained from all participants.

**suPAR assessment**. Plasma suPAR levels were measured in the DBDS cohort and two validation cohorts (Dunedin and E-Risk) using the CE/IVD-approved suPARnostic AUTO Flex ELISA (ViroGates A/S, Birkerød, Denmark) following the manufacturer's instructions. The suPARnostic assay utilizes two monoclonal antibodies: a capture antibody directed towards the $D_{III}$ subunit and a detection antibody against the $D_{II}$ subunit. Full-length suPAR ($D_I D_{II} D_{III}$) may be cleaved into $D_I$ and $D_{II} D_{III}$, and the assay captures free full-length suPAR ($D_I D_{II} D_{III}$) as well as the suPAR fragment ($D_{II} D_{III}$) but not the $D_I$ fragment. The $D_I D_{II} D_{III}$ full-length suPAR molecule can bind urokinase plasminogen activator (uPA) and $D_I D_{II} D_{III}$/uPA complexes will not be detected in the suPARnostic assay[68]. suPAR levels were measured in 14,367 participants in DBDS, 1444 in E-Risk, and 837 in Dunedin. suPAR levels were measured at age 18 in the E-Risk Study, as previously described[24] whereas suPAR levels were measured at age 38 in the Dunedin Study, as previously described[69].

For the Icelandic cohort, suPAR is one of the plasma proteins measured using the SOMAscan platform as described above.

**Genotyping and imputation**. Genotyping and imputation of the 110,000 DBDS Genetic Cohort is described in Hansen et al.[65]. Briefly, DNA purification is performed from the whole blood samples and immediately stored at $-20\,^{\circ}C$. The samples were genotyped using the Global Screening Array by Illumina, which includes >650,000 variants with custom chip content optimized for comparison with the Illumina Omni Express chip. All genotype data are processed simultaneously for genotype calling, quality control, and imputation. Quality control was conducted in both populations, including using a minimum allele count of 5, the exclusion of individuals or variants with more than 10% missingness, and individuals deviating more than three standard deviations (SDs) from the population heterozygosity (correcting for individuals carrying large copy number variations, >100 Kbp). We performed imputation using a reference panel backbone consisting of (1) UK 1 KG phase 3 and HapMap reference to predict non-genotyped variants with minor allele frequency (MAF) > 1%, and (2) an in-house dataset consisting of $N > 6000$ Danish whole-genome sequences to improve the prediction of variations with a MAF down to around 0.01%.

The process used to whole-genome sequence the 49,708 Icelanders, as well as the subsequent imputation, has been described in previous publications[70,71]. Briefly, we sequenced the whole genomes of 49,708 Icelanders using Illumina technology to a mean depth of at least 10× (median 32×). SNPs and indels were identified and their genotypes called using joint calling with Graphtyper[72]. In total, 166,281 Icelanders were genotyped using Illumina SNP chips and their genotypes were phased using long-range phasing[73]. All sequenced individuals were also chip-typed and long-range phased, providing information about haplotype sharing that was subsequently used to improve genotype calls. Genotypes of the 32 million high-quality sequence variants were imputed into all chip-typed Icelanders. Using genealogic information, the sequence variants were also imputed into relatives of the chip-typed further increasing the sample size for association analysis and the power to detect associations. All the variants tested had imputation information over 0.8.

For the two validation cohorts, we used Illumina HumanOmni Express BeadChip arrays (Illumina CA, USA) to assay common single nucleotide polymorphism (SNP) variation in the genomes of participants of the E-Risk and Dunedin studies, as previously described[74]. The resulting database was restricted to SNPs called successfully in >98% of each cohort and in Hardy-Weinberg equilibrium ($p > 0.001$). Additional SNPs were imputed using the IMPUTE2 software (version 2.3.1, https://mathgen.stats.ox.ac.uk/impute/impute_v2.html)[75] and 1000 Genomes version-3 reference panel[76]. Imputation was conducted on autosomal SNPs appearing in dbSNP (version 140; http://www.ncbi.nlm.nih.gov/SNP/)[77] that were called in >98% of each sample. Invariant SNPs were excluded. Prephasing and imputation were conducted using a 50 M base-pair sliding window. The resulting genotype database included genotyped SNPs and SNPs imputed with a 90% probability of a specific genotype among European-descent E-Risk members ($n = 1999$ children in 1011 families) and among the non-Maori members of the Dunedin cohort ($n = 918$).

**Twin/heritability analysis**. To test the genetic contribution to suPAR levels at age 18 in the E-Risk study, we used a univariate twin model comparing correlations between MZ and DZ twins to decompose the phenotypic variation in plasma levels of suPAR into additive genetic, shared environmental, and unique environmental components. We used Mplus Version 7.4 (Muthen & Muthen, Los Angeles, CA) for the analysis. We additionally calculated SNP heritability based on the Icelandic cohort. We estimated the narrow-sense heritability of suPAR with LD score

regression[78], using an LD score map calculated with high-quality markers from the Icelandic population.

**GWAS statistical analysis**. The suPAR measurements were each rank-based inverse normal transformed to a standard normal distribution (separately for each sex) and adjusted for age using a generalized additive model. A linear mixed model implemented by BOLT-LMM[79], was used to test for association between sequence variants and suPAR levels, assuming an additive genetic model. Thirty-five million variants are tested in Iceland, while 26 million variants are tested in Denmark.

BOLT-LMM accounts for cryptic relatedness and population stratification[79], and we additionally used LD score regression to account for distribution inflation in the dataset due to cryptic relatedness and population stratification[78]. The Danish and Icelandic datasets were combined using a fixed-effect inverse-variance weighted meta-analysis, allowing the populations to have different frequencies for alleles and genotypes but assuming them to have a common effect. Heterogeneity in effect estimates was assessed using a likelihood-ratio test. Effects are given in units of SDs. In total 40 million variants are tested either in Iceland or Denmark, of which 21 million variants are tested in both datasets. Rare variants may therefore be present only in one discovery study.

We accounted for multiple testing by means of a weighted Bonferroni correction, taking into account the higher prior probability of association of certain variant annotations while controlling the family-wise error rate (FWER) at 0.05[31]. The method has been described previously[31] and results in stricter multiple testing correction than the commonly used threshold of $5 \times 10^{-8}$ (which would not control FWER at 0.05 given that 40 million markers were tested) while being more powerful than simply correcting for 40 million tests using a fixed threshold of $0.05/40,000,000 = 1.25 \times 10^{-9}$. The resulting significance thresholds were $2.0 \times 10^{-7}$ for high-impact variants (including stop-gained, frameshift, splice-acceptor, or splice-donor variants, $N = 11,723$), $4.0 \times 10^{-8}$ for 'moderate-impact' variants (including missense, splice-region variants, and in-frame indels, $N = 202,336$), $3.7 \times 10^{-9}$ for 'low-impact' variants (including upstream and downstream variants, $N = 2,896,354$), and $6.1 \times 10^{-10}$ for the 'lowest-impact' variants (including intron and intergenic variants, $N = 37,239,641$). To identify whether several variants in a single locus are independently associated with suPAR, we performed conditional analysis using Icelandic individual-level data, where LD data are available from the same population. This is in contrast with methods such as GCTA which use summary-level data only. The variants' predicted genes are based on their actual position within a given gene or the closest gene (from Ensembl variant effect predictor information[80]).

For previous associations and functions of our suPAR-associated variants and genes, we performed manual searches using PubMed (pubmed.ncbi.nlm.nih.gov). For the variant's associated gene, we used the UniProt Protein Knowledge Base (UniProtKB) to find the function and annotated GO biological processes listed for each gene (www.uniprot.org/uniprot)[81].

For the replication phase, additive genetic association tests between suPAR levels and each of the significant variants were performed using the R package "SNPassoc". The model suPAR ~variant + sex + PCs1-10 was employed, where variant genotypes were coded as number of risk alleles (0,1,2). For the E-Risk twin study, the analyses employed generalized estimating equation (GEE) linear regression models using the R package 'gee', taking into account the clustering of the twins within families.

**Comparison of genetic variants' effects for suPAR unadjusted vs adjusted for smoking**. As suPAR levels have strong associations with smoking, we investigated whether smoking status would affect the outcome of the suPAR GWAS results. Smoking status was available for 30,469 of the 35,559 individuals with suPAR measurements in the Icelandic data. This includes 11,093 non-smokers and 19,376 smokers, where "smoker" was defined as "ever smoker". To assess if the inclusion of smoking as a covariate could have an effect on the GWAS outcomes, we performed two GWASs: (1) the 30,469 individuals with available information on smoking status, unadjusted for smoking; and (2) the same 30,469 individuals, adjusted for smoking. A test of difference (heterogeneity) in the GWAS results was performed.

**Biological Network Gene Ontology Analysis**. To quantitatively investigate biological processes associated with each gene, we used the Biological Network Gene Ontology (BiNGO) bioinformatics tool[44] to assess whether there are GO terms biological processes that are statistically overrepresented in our set of 12 suPAR-associated genes. The BiNGO tool uses GO terms from the Gene Ontology database (www.geneontology.org) and calculates the P values for overrepresented biological processes in our set of genes using the hypergeometric test. This takes into account both the total number of genes from the input dataset and the total number of genes for the specific GO term biological process. A total of 14,306 genes were available in the tool's reference set. The Benjamini-Hochberg (false discovery rate) correction is calculated to control for multiple testing, where only significantly overrepresented GO term processes with corrected $P < 0.05$ were considered.

**PheWAS statistical analysis**. To gain further insights into the possible functional and regulatory role of our newly identified variants, phenotype-wide association analyses (pheWASs) were conducted. pheWASs using suPAR PRSs as the exposure were performed to investigate whether the combined effect of these suPAR-associated genetic variations were associated with specific phenotypes. PRSs were calculated for each individual in the Icelandic population based on the summary statistics of the DBDS suPAR GWAS, and PRSs for suPAR were calculated for each of the 500,000 UK Biobank subjects based on the meta-analysis summary statistics (Denmark + Iceland). Briefly, to generate the suPAR-PRS for the UK Biobank sample we used 630,000 informative variants across the genome and constructed locus allele-specific weightings by applying LDpred to the summary data from the subset meta-analysis GWAS[82]. Constructing individual weightings, we were able to calculate an aggregated score of suPAR in all included individuals. Subsequently, we assessed the impact of suPAR-PRS on 63,609 traits (binary and quantitative) using a Bonferroni significance threshold of $P < 7.86 \times 10^{-7}$. More specifically, a pheWAS was performed in a comprehensive phenotype dataset within the Icelandic population consisting of 14,493 case/control phenotypes and 28,389 quantitative phenotypes, as well as pheWAS in UK Biobank with 15,120 case/control phenotypes and 5609 quantitative phenotypes[83]. In addition, single-variant pheWASs were performed in the same datasets using the same Bonferroni significance threshold.

**Mendelian randomization analyses for suPAR vs. the identified pheWAS findings**. We performed Mendelian randomization analyses for suPAR and the identified pheWAS findings using phenotypes available in the Icelandic population cohort. The analyses were performed using the R Package "MendelianRandomization"[84], using the inverse variance weighted (IVW) and MR-Egger methods. The MR-PRESSO global test[85] was additionally used to detect possible outliers and remove them. In the instances where outliers were found, the outliers were removed and the IVW method was reutilized. The instrumental variables (IVs) used are the variants we report significant for suPAR, i.e., the lead variants.. Mendelian randomization for suPAR and 10 phenotypes were performed, including type 1 diabetes, autoimmune diseases as a general category, obesity, rheumatoid arthritis, B12, fasting glucose, alkaline phosphatase, and potassium.

**Cis-eQTL analysis of the suPAR-associated variants**. We tested if our variants were in high LD ($r^2 > 0.8$) with one more top cis-eQTL based on various tissues and 17 sources including GTEx and Icelandic data (See Supplementary Data 15 for list of sources). For the Icelandic data, RNA sequencing and estimation of the association between sequence variants and gene expression have been described in a recent publication[86].

**Genetic correlation between suPAR and CRP**. The genetic correlation between suPAR and CRP was calculated using suPAR from our Iceland/Denmark meta-analysis and CRP from UK biobank. We additionally investigated the overlap of findings between the suPAR-associated variants identified in our study and CRP-associated variants from the GWAS-catalog (https://www.ebi.ac.uk/gwas/). CRP-associated variants reported in the GWAS-catalog were checked for overlap with our suPAR-associated variants + LD-classes (all variants with $r^2 > 0.80$).

**Reporting summary**. Further information on research design is available in the Nature Research Reporting Summary linked to this article.

## Data availability

The suPAR meta-analysis summary statistics will be made available at https://www. decode.com/summarydata/.

Sequence variants passing GATK filters have been deposited in the European Variation Archive, accession number PRJEB15197.

For information on further access to data included in the meta-analysis, please contact the following authors of the respective cohorts: Hreinn Stefansson for data from the Icelandic cohort (hreinn.stefansson@decode.is) and Sisse Rye Ostrowski for data from the Danish Blood Donor Study (Sisse.Rye.Ostrowski@regionh.dk). The Dunedin Study data and E Risk study data are not publicly available but are available on request by qualified scientists. Requests require a concept paper describing the purpose of data access, ethical approval at the applicant's institution, and provision for secure data access. Secure access is possible on the Duke University, Otago University, and King's College London campuses. For UK Biobank please register on https://bbams.ndph.ox.ac.uk/ams/ and apply for the data through there.

## Code availability

Variants in the Icelandic and Danish cohorts were imputed using software developed at deCODE genetics based on the IMPUTE HMM model[87] as previously described[88]. A linear mixed model implemented by BOLT-LMM[79] was used to test for association between sequence variants and suPAR levels.

We used publicly available software (URLs listed below) in conjunction with the above described algorithms in the sequencing processing pipeline (Whole-genome sequencing, Association testing, RNA-seq mapping, and analysis):

BWA 0.7.10 mem, https://github.com/lh3/bwa
GenomeAnalysisTKLite 2.3.9, https://github.com/broadgsa/gatk/
Picard tools 1.117, https://broadinstitute.github.io/picard/
SAMtools 1.3, http://samtools.github.io/
Bedtools v2.25.0-76-g5e7c696z, https://github.com/arq5x/bedtools2/
Variant Effect Predictor https://github.com/Ensembl/ensembl-vep
BOLT-LMM https://data.broadinstitute.org/alkesgroup/BOLT-LMM/downloads/"
IMPUTE2 v2.3.1 https://mathgen.stats.ox.ac.uk/impute/impute_v2.html
dbSNP v140; http://www.ncbi.nlm.nih.gov/SNP/
LD Score Regression software; https://github.com/bulik/ldsc
BiNGO v3.0.3 https://www.psb.ugent.be/cbd/papers/BiNGO/Download.html
Cytoscape v3.7.1 https://cytoscape.org/download.html
We used R extensively to analyze data and create plots.

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

## Acknowledgements

We thank participants in all the included cohorts. Similarly, we thank personnel employed in all blood banks across Denmark for making DBDS inclusion a part of their work routine. This study was supported by the Danish Council for Independent Research (09-069412 and 0602-02634B), Aase og Ejnar Danielsens Fond, AP Møller Fonden, and the Danish Regions (02/2611). The DBDS genetic infrastructure was supported by the Novo Nordic Foundation (NNF17OC0027594). S.B. reports grants from Innovation Fund Denmark, grants from Novo Nordisk Foundation during the conduct of the study; and personal fees from Intomics A/S and Proscion A/S, outside the submitted work. S.B. and K.B. report on behalf of Novo Nordisk Foundation Center for Protein Research, University of Copenhagen, that the following grants supported the study: The Novo Nordisk Foundation (NNF14CC0001 and NNF17OC0027594) and The Innovative Medicines Initiative 2 Joint Undertaking under grant agreement no. 115881 (RHAP-SODY). L.J.H.R. is supported by an international postdoctoral fellowship from the Lundbeck Foundation (grant no. R288-2018-380). The E-Risk Study is funded by the Medical Research Council (UKMRC grant G1002190). Additional support was provided by the National Institute of Child Health and Human Development (grant HD077482) and by the Jacobs Foundation. The authors are grateful to the Study members and their families for their participation. Our thanks CACI, Inc., and to members of the E-Risk team for their dedication, hard work, and insights. We thank the Dunedin Study members, Unit research staff, and Study founder Phil Silva. This research was supported by US-National Institute on Aging grants AG032282 and UK Medical Research Council grant MR/P005918/1. The Dunedin Multidisciplinary Health and Development Research Unit is supported by the New Zealand Health Research Council Programme Grant (16-604), and the New Zealand Ministry of Business, Innovation, and Employment (MBIE).

## Author contributions

J.D. contributed to the conception, design, analysis, interpretation of data and drafted the work. E.F., M.K.M., L.J.H.R., and K. Sugden contributed to the design, analysis, interpretation of data and participated in the revision of the first manuscript draft. The DBDS Genomic Consortium contributed to the acquisition of data. G.T., V.T., and M.F. contributed to the acquisition and the analysis of data. S.H.L., L.S., and B.G. contributed to the analysis of data. L.W.T., K.S.B., S.R.O., E.S., C.E., O.B.P., T.F.H, K.B., S.B., R.P., L.A., A.C., T.E.M., and D.G. contributed to the acquisition, interpretation of data and participated in the revision of the first manuscript draft. J.E.O., H.S., K. Stefánsson, and H.U. participated in the acquisition of data, design of the study, interpretation of data, and to the revision of the first manuscript draft. All authors approved the submitted version of the manuscript.

## Competing interests

The authors E.F., G.T., M.F., M.K.M., V.T., S.H.L., L.S., B.G., D.G., H.S., and K. Stefánsson who are affiliated with deCODE genetics/AMGEN, declare competing interests as employees. J.E.O. is named inventor on patents on suPAR owned by Copenhagen University Hospital Hvidovre, Denmark. J.E.O. is co-founder, shareholder, and board

member of ViroGates A/S, Denmark, the company that developed the suPARnostic ELISA. ViroGates A/S had no role in the study design, data collection, analysis, or interpretation of study findings, and/or in the decision to submit the manuscript for publication. The remaining authors declare no competing interests.

## Additional information

## DBDS Genomic Consortium

**Denmark** Steffen Andersen[16], Karina Banasik [9], Søren Brunak [9], Kristoffer Sølvsten Burgdorf[1], Christian Erikstrup [6], Thomas Folkmann Hansen [8,9], Gregor Jemec[17], Poul Jennum[18], Rene Kasper Nielsen[19], Mette Nyegaard[20], Helene Martina Paarup[21], Ole Birger Pedersen [7], Mikkel Petersen[22], Erik Sørensen[1], Henrik Ullum[1] & Thomas Werge[23]

**Iceland** Daniel Gudbjartsson[2,15], Kari Stefansson[2,5], Hreinn Stefánsson[2] & Unnur Þorsteinsdóttir[2]

[16]Department of Finance, Copenhagen Business School, Copenhagen, Denmark. [17]Department of Clinical Medicine, Zealand University hospital, Roskilde, Denmark. [18]Department of Clinical Neurophysiology, University of Copenhagen, Copenhagen, Denmark. [19]Department of Clinical Immunology, Aalborg University Hospital, Aalborg, Denmark. [20]Department of Biomedicine, Aarhus University, Aarhus, Denmark. [21]Department of Clinical Immunology, Odense University Hospital, Odense, Denmark. [22]Department of Clinical Immunology, Aarhus University Hospital, Aarhus, Denmark. [23]Institute of Biological Psychiatry, Mental Health Centre Sct. Hans, Copenhagen University Hospital, Roskilde, Denmark.

