## [Peer Review File · Communications Biology]

Reviewers' comments:

Reviewer #1 (Remarks to the Author):

The paper reports the genetics of suPAR, the soluble form of the membrane-bound receptor urokinase-type plasminogen activator receptor (uPAR) and an inflammatory marker that has been associated with mortality and chronic diseases. The study used a twin study to examine the heritability and shared and nonshared environment effects, and two large European studies for a genome-wide association discovery of loci associated with suPAR. Main findings are 11 associated loci (13 variants), of which 8 were common variants and the remaining were rare/low frequency. Validation was performed in two additional studies for only the significantly associated variants. Identifying genetic variants for suPAR is of interest given its role in inflammation and chronic diseases. There are several aspects of the manuscript that need further details and/or clarification as listed below.

Thresholds for discovery and replication should be clearly stated in the results section. Table 2 shows that some associations are only nominal in replication samples.

Minimum allele count used in analyses are not mentioned.

Potential differences in suPAR assay methodology across studies needs to be discussed as part of the harmonization. A direct comparison of the two assays in same samples would be preferred. Include in the main text the ancestry for the validation studies: Dunedin Multidisciplinary Health and Development Study (Dunedin) cohort and the E-Risk cohort as well as the number of participants in these samples (Table 2). Table 2, include chromosome:position for each variant. For each identified variant in Table 2, coding and non-coding, the information on protein impact should be moved from supplementary material to the main text.

The section "summary of the suPAR associated loci" (page 7), describe only variants that replicated.

Page 12, PRS analyses, it looks like 630,000 variants were used to create the PRS, but they explain less than 1% of the variance of the trait. I am not sure how to interpret the results from this weak PRS for suPAR.

Discussion, page 13, related to variant in and near PLA2R1 that increases suPAR levels (rs3828323) and PRS that is associated with PLA2R1 serum levels. This is interesting given the known causal relationship of PLA2R1 with membranous nephropathy. A question is if participants with chronic kidney disease and/or glomerulopathies were excluded from the discovery. Further examination of this locus and variants identified in a recent GWAS of membranous nephropathy may add strength to the manuscript.

What is the overlap of findings with CRP, another inflammatory circulating protein?

Methods:

The Icelandic study included several datasets such as a cohort of cancer patients. How the different samples were handled, including different sampling from the population, differences in genotype platform and so on.

What were the criteria to consider a significant association? Did you select variants that were significant in one or other discovery datasets or only from meta-analyses of discovery GWAS studies? Given differences in imputation reference panels for discovery studies (WGS in the Iceland samples and 1000G in the other cohort), are the rare variants present only in one discovery study? Include the number of variants tested in the discovery for each study.

Related to above, most of the variants shown in Table 2 are within a gene, which suggest that there is an over-representation of imputed data in coding region in the discovery and/or the results are driven by one study with more dense imputation.

It is not clear how relatedness and population stratification were accounted for in statistical analyses in the Iceland study. Also confusing is the description of thresholds for significance based on variant impact which suggest that the genotypes were predominantly exonic variants.

Reviewer #2 (Remarks to the Author):

This is an interesting GWAS study of an inflammatory marker named su-PAR. To get further insights into the cause of chronic inflammation, it is of highly interest to study the genetics of different inflammatory markers and therefore I think this study is of high importance. The

statistical methods are sound, the manuscript reads well.

I have a few comments/suggestions to the authors that I would like to see addressed:

1. I miss genome-wide genetic correlation analysis, especially with CRP (2018 paper). I recommend to use ld score regression.
2. I miss conditional analysis using GCTA. Are more variants in a single locus associated? This may also improve variance explained.
3. I would recommend the authors to look into potential eQTL effects of the variants found in existing databases. This would provide further info with regards to causal genes.
4. The authors list in the text the numbers that are in the table 1. Please omit this text, the readers can find it in the tables. I would suggest to explain gene functions in a separate box.
5. I would also like to see proper mendelian randomization analysis for the identified PHEWAS findings.
6. Methods: Which software was used for the GWAS meta-analysis? Please mention.
7. Danish cohort methods: random selection patients for su-PAR measurements?
8. E-Risk study methods: first sentence about Dunedin study.

Symen Ligthart, Erasmus University Medical Center, Rotterdam, the Netherlands

Reviewer #3 (Remarks to the Author):

Comments to the authors

The authors have presented an in-depth study, investigating the potential genetic role in the variation of suPAR levels as well as its potential role in relation to risk of disease, such as inflammation. suPAR, which is normally used as a non-specific biomarker for chronic inflammation, have been extensively studied in relation to lifestyle factors, however the genetics behind changes in marker is lesser known. The authors here present two independent GWAS in an Icelandic and a Danish cohort, which were then meta-analysed together. Additionally, they validate their results in a UK-based cohort as well as a New Zealand-based. They present possible candidate genes based on genomic location and gene ontology / biological processes analysis. Lastly a heritability analysis is performed as well as a pheWAS for computed polygenic risk scores as well as for the independently associated variants from the meta-analysis. This is a well-written article, and it contributed to further knowledge within the field.

However, there are some points that could improve even more with clarification or further discussion/commenting.

General questions

It is mentioned in the introduction that life factors associated with suPAR have been extensively studied, especially smoking. In the main analyses, only sex is used as a covariate. Do the authors have access to lifestyle factors such as smoking status? Would this and other potential life style factors affect the outcome of the primary results in the GWASs? Is there a reason principal component were only used in the replication analyses?

How do the heritability estimates look in the other non-twin-based cohorts? Is there a possibility to calculate SNP heritability in the Icelandic and Danish cohort? Would it differ much from the PRSs, i.e. explain more of the difference between the variance explained by the PRS and the genetic additive effects explained in the MZ twin setting? Would this explain some of the gap between the MZ estimates and the PRSs?

When talking about the results and discussing further downstream analyses, only the variants from the meta-analysis considered. As of now, it is a bit unclear in the text whether the 11 loci are completely overlapping between the Icelandic and the Danish cohort or if there were cohort-specific signals. It is clear in S Table 1, the cohort section, that rs755902185 was not picked up in the DBDS for example, but I think it would be neat to present it also in the main text for clarification.

The suPAR measurements have been measured at different time points in the cohorts (at 18yrs in E-Risk and 38yrs in Dunedin and not stated in the rest). It is stated that suPAR is stable in terms of minimal circadian changes, but does the time of measurement affect further analysis? Does the age component in the regression capture any possible variation due to age (if present)?

Figure 1: The Manhattan plot is a bit confusing. The labelled variants are the independently associated variants at the 11 loci, however they are rarely the lead variant at respective locus (i.e. lowest p value). How are the independent variants at each locus defined?

How many phenotypes are overlapping between the Icelandic cohort and in UK Biobank? Is there a possible reason why the UK Biobank based pheWAS does not yield as many as or similar results as the Icelandic based?

Minor clarifications.

In general, I think it would be helpful to add footnotes to the tables, as with S. Table 4, making them even more clear to follow; e.g., adding a footnote for nAff and nCon in S. Table 6, which I assume to be number of affected and number of controls respectively.

For better comparison, it would be good if the allele frequencies (effect and minor) are stated as a fraction rather than a percentage in the supplementary tables (S Table 1-2,8-11) to match the tables in the main article or vice versa.

Line 235 is a bit confusing, "to investigate the combined effect of these suPAR-associated genetic variations", assuming it is the combined effect of all significantly associated variants in the GWASs. The sentence might benefit from being rephrased.

Line 374-375: "Environmental Risk (E-Risk) Longitudinal Twin Study: suPAR levels were measured at age 18 in the Dunedin study as previously described": I assume Dunedin is wrongly stated and should be E-Risk?

What software was used for the heritability and GWAS analyses? And why different software for all different analyses?

Line 89-92: Are the additive effects + shared environment + nonshared environment supposed to add up to 100%?

Response to Reviewers for the suPAR GWAS Meta-analysis Manuscript

Referee #1: GWAS, nephrology

	The paper reports the genetics of suPAR, the soluble form of the membrane-bound receptor urokinase-type plasminogen activator receptor (uPAR) and an inflammatory marker that has been associated with mortality and chronic diseases. The study used a twin study to examine the heritability and shared and nonshared environment effects, and two large European studies for a genome-wide association discovery of loci associated with suPAR. Main findings are 11 associated loci (13 variants), of which 8 were common variants and the remaining were rare/low frequency. Validation was performed in two additional studies for only the significantly associated variants. Identifying genetic variants for suPAR is of interest given its role in inflammation and chronic diseases. There are several aspects of the manuscript that need further details and/or clarification as listed below.	We thank the reviewer for their thorough review which has provided us with relevant critical feedback. We have responded to their comments in a point-by-point manner below.
#	Referee	Response to reviewer
1	Thresholds for discovery and replication should be clearly stated in the results section. Table 2 shows that some associations are only nominal in replication samples.	We appreciate the reviewer’s suggestion to state the thresholds for the meta-analysis and replication phase more clearly in the results section. We wish to emphasise that our criteria for statistical significance is based on a weighted Bonferroni adjustment, using the enrichment of sequence annotations among association signals as weights. Several thresholds are therefore used depending on the variant type. For the replication phase, we use $P < 0.05$ as the threshold. We have now added this to the Results section on page 5, lines 99-105: “We performed a meta-analysis of the two GWASs (N=47,736) and employed a weighted Bonferroni adjustment to determine statistical significance as previously described (PMID: 26854916). The P-value significance thresholds were 2.0×10^{-7} for high-impact variants (including stop-gained, frameshift, splice-acceptor, or splice-donor variants, N=11,723), 4.0×10^{-8} for ‘moderate-impact’ variants (including missense, splice-region variants, and in-frame indels, N=202,336), 3.7×10^{-9} for ‘low-impact’ variants (including upstream and downstream

		variants, N=2,896,354), and 6.1×10^{-10} for the ‘lowest-impact’ variants (including intron and intergenic variants, N=37,239,641).” On page 8, lines 145-146: “P-values less than 0.05 were considered statistically significant in the validation phase.”
2	Minimum allele count used in analyses are not mentioned.	Both the Icelandic and Danish analyses use a minimum allele count of 5. This has been added to Methods (“Genotype and Imputation”). On page 22, lines 491-492: “Quality control was conducted in both populations, including using a minimum allele count of 5 [...]”
3	Potential differences in suPAR assay methodology across studies needs to be discussed as part of the harmonization. A direct comparison of the two assays in same samples would be preferred.	We understand the reviewer’s comment that a discussion on potential differences between the two different suPAR detection methods should be made. The important comparison in this genetic study of suPAR is the comparison of the results of the GWAS findings between the populations measured with these two different assays. When comparing the GWAS for the Icelandic (proteomics-based assay) and Danish (ELISA) populations there is high concordance; only one marker shows a significant p for heterogeneity (Qp), rs71311394 (3’prime UTR variant in ST3GAL6); P=0.003. This variant has genome-wide significant association in both populations but the effect size is larger in the Danish population compared to the Icelandic (0.21 vs 0.11). Given this high degree of similarity in the GWAS findings we feel that a direct comparison between the assays would not add any further genetic insights into our findings. We have added the following discussion on pages 17, lines 373-381: “Two different suPAR detection methods were used in our study; a proteomics-based assay in the Icelandic population and ELISA in the Danish population. Although a direct comparison between the two different assays using the same samples was not possible, a comparison of the results of the GWAS findings between the Danish and Icelandic population cohorts was performed, of which the results showed high concordance. Only one variant, the rs71311394 variant in ST3GAL6, shows evidence of heterogeneity (P=0.003). However, direction of effects for rs71311394 are consistent between the two populations (Effect_{Ice}=0.11 vs. Effect_{DK}=0.21) and the association with suPAR levels is genome-wide significant in both populations (P_{Ice}=5.3×10^{-12} vs. P_{DK}=2.1×10^{-18}). Given this high degree of similarity in the GWAS findings we believe that a direct comparison between the assays would not add any further genetic insights into our findings.”
4	Include in the main text the ancestry for the validation studies: Dunedin Multidisciplinary Health	Thank you for these valuable suggestions. We have added ancestry in the main text as well as number of participants in these samples. We have also added chromosome and position

	and Development Study (Dunedin) cohort and the E-Risk cohort as well as the number of participants in these samples (Table 2). Table 2, include chromosome:position for each variant.	columns for each variant in table 2. On page 7, lines 140-144: “We used a sample consisting of 837 individuals of white European-descent non-Maori descent from The Dunedin Multidisciplinary Health and Development Study (Dunedin) cohort, of which eight of the 13 variants were available for replication. A sample of 1,444 E-Risk members of white European-descent was also used as a validation cohort, of which six of the 13 variants were available.”
5	For each identified variant in Table 2, coding and non-coding, the information on protein impact should be moved from supplementary material to the main text.	We understand the reviewer’s suggestion, however we would like to note that Table 1 already provides this information in the main text, along with other variant information and summary statistics, and the focus of Table 2 is the validation results. As Table 2 features the same 13 variants as shown in Table 1, we do not believe it would be necessary to add Variant Type column in this table, as this can already be found in Table 1. Similarly, as we have already implemented the reviewer’s request of adding chromosome and position to Table 2 (both of which also can be found in Table 1), we do not wish to unnecessarily expand Table 2 further. However, if the reviewer insists, we will happily extend Table 2.
6	The section “summary of the suPAR associated loci” (page 7), describe only variants that replicated.	We understand the reviewer’s suggestion regarding describing only the variants that replicated. We do believe however that it is relevant to include a brief summary of all 11 loci that were discovered in the meta-analysis. With the exception of variant rs7952602 in ST3GAL4 , all other variants that were available in the validation cohorts were confirmed in at least one of the two cohorts, with all available variants showing effect estimates in the same direction, with the exception of rs71311394 (ST3GAL6). Due to the replication cohorts' smaller N, five suPAR-associated variants from the meta-analysis were not available for replication (especially for variants with very low allele frequencies or because of variants not surviving quality control). We believe these variants are still of interest to be described briefly in this section despite their unavailability for replication. We would also like to note as in comment 3 that the variants were tested for heterogeneity between the Danish and Icelandic cohorts, of which the results showed remarkable consistency between the two large and independent cohorts.
7	Page 12, PRS analyses, it looks like 630,000 variants were used to create the PRS, but they explain less than 1% of the variance of the trait. I am not sure how to interpret the results from this weak PRS for suPAR.	We agree with the reviewer’s observation that the current variance explained by our PRSs is low. We acknowledge this finding in the Discussion and that larger meta-analyses with larger cohorts may explain more of the variance. Missing heritability is a frequently reported situation in GWASs and so we do not believe this can be considered a shortcoming of our study. Page 18, lines 388-195: “However, it must be noted that our PRSs based on the Icelandic cohort’s suPAR GWAS only explain 1.00% of suPAR variance when predicting into the Danish cohort, and

		0.94% variance of suPAR when predicting from the Danish cohort's suPAR GWAS into the Icelandic cohort. As our univariate twin model estimated that additive genetic effects account for 60% of the variation in suPAR levels, we are unable to account for much of the heritability of suPAR, a situation frequently reported in GWASs of other phenotypes (PMID: 27241833). Larger studies will likely enable the discovery of more variants that explain some of this missing heritability, as well as future studies focusing on other forms of genetic variation such as copy number variants.”
8	Discussion, page 13, related to variant in and near PLA2R1 that increases suPAR levels (rs3828323) and PRS that is associated with PLA2R1 serum levels. This is interesting given the known causal relationship of PLA2R1 with membranous nephropathy. A question is if participants with chronic kidney disease and/or glomerulopathies were excluded from the discovery. Further examination of this locus and variants identified in a recent GWAS of membranous nephropathy may add strength to the manuscript.	We agree with the reviewer that this locus is interesting. We have looked at membranous nephropathy variants at this locus, based on two studies: “Single nucleotide polymorphisms in the phospholipase A2 receptor gene are associated with genetic susceptibility to idiopathic membranous nephropathy” (PMID 20805699 – reference #34 from our manuscript) & “The genetic architecture of membranous nephropathy and its potential to improve non-invasive diagnosis” (PMID: 32231244). We have added the findings in the discussion as mentioned below. We have not excluded individuals with chronic kidney disease from the cohorts, although in the Danish DBDS cohort it is a requirement that blood donors must be generally healthy, including no kidney diseases. Therefore there should not be any participants with chronic kidney disease at the time of suPAR measurement in the DBDS cohort. We have added the following to the Discussion. Page 15, lines 332-338: “Two studies on membranous nephropathy have identified associated genetic variants, of which one found association with two variants: rs3828323 and rs3577198232, and the other study identified had rs17831251 as the lead associated variant⁵⁰. The two variants that are not reported in our study, i.e. rs35771982 and rs17831251 associate with suPAR in our meta-analysis but these associations do not hold up after adjusting for our lead variant ($r^2=0.32$ and $r^2=0.19$ respectively). This would indicate that the signals reported in these previous membranous nephropathy genetic studies at the PLA2R1 locus are the same as for suPAR in our study”.
9	What is the overlap of findings with CRP, another inflammatory circulating protein?	We have checked for CRP variants from the GWAS-catalog for our suPAR-associated variants + LD-classes (all variants with $r^2>0.80$). The only overlapping variant is the FUT2 variant (rs601338), where the same variant was reported in PMID: 31900758. The rs601338 variant has also been reported for several other phenotypes, see https://www.ebi.ac.uk/gwas/variants/rs601338. We have additionally looked up all the genes that we report to see if they have been reported with CRP, of which the only overlap we find in addition to FUT2 is ABO, where there is a variant at chr9:133266942 (rs643434) reported to associate

		with CRP in the paper PMID: 30388399. This variant has $r^2=0.17$ with “our” ABO variant. We have additionally performed a genetic correlation analysis between suPAR and CRP. The genetic correlation between suPAR and CRP was calculated to be 0.2351 (SD=0.0533, $P=1.03e-5$), using suPAR from Iceland/Denmark meta-analysis and CRP from UK biobank. These additional analyses that have been requested have been added to the manuscript. In Methods on page 27, lines 615-620: “The genetic correlation between suPAR and CRP was calculated using suPAR from our Iceland/Denmark meta-analysis and CRP from UK biobank. We additionally investigated overlap of findings between the suPAR-associated variants identified in our study and CRP-associated variants from the GWAS-catalog (https://www.ebi.ac.uk/gwas/). CRP-associated variants reported in the GWAS-catalog were checked for overlap with our suPAR-associated variants + LD-classes (all variants with $r^2>0.80$).” In Results, page 14, lines 297-307: We performed a genetic correlation analysis between suPAR and CRP. The genetic correlation between suPAR and CRP was calculated to be 0.2351 (SD=0.0533, $P=1.03\times 10^{-5}$), using suPAR measurements from the Icelandic/Danish meta-analysis and CRP measurements from UK Biobank. We have additionally checked for CRP variants from the GWAS-catalog for our suPAR-associated variants and their LD-classes (all variants with $r^2>0.80$). Only the FUT2 variant (rs601338) was found to be an overlapping variant, where the same variant was reported in Han et al (PMID: 30388399). We have additionally searched all the genes that we report to see if they have been reported with CRP, of which the only overlap we find in addition to FUT2 is ABO, where there is a variant at chr9:133266942 (rs643434) reported to associate with CRP in the CRP GWAS meta-analysis by Ligthart, et al (PMID: 30388399). This variant has $r^2=0.17$ with the ABO variant found in our study.
10	The Icelandic study included several datasets such as a cohort of cancer patients. How the different samples were handled, including different sampling from the population, differences in genotype platform and so on.	The majority of the samples (52%) were collected at the same site, in conjunction with the Icelandic Cancer Project (ICP). In the ICP, all prevalent and newly diagnosed Icelandic cancer cases and their relatives were invited to participate in a comprehensive study of cancer, along with a control population, randomly selected from the National Registry. That population might be slightly enriched for cancer and other diseases. The remaining samples (48%) were collected as part of various genetic programs at deCODE genetics, Reykjavík, Iceland. The median collection date for samples collected in conjunction with ICP was July 1st 2002, whereas the median

		collection date for other samples was May 15th 2015. Genotyping is done in the same way for both cohorts, as described in methods. Page 19, lines 423-427: “In the ICP, all prevalent and newly diagnosed Icelandic cancer cases and their relatives were invited to participate in a comprehensive study of cancer, along with a control population, randomly selected from the National Registry. The median collection date for samples collected in conjunction with ICP was July 1st 2002, whereas the median collection date for other samples was May 15th 2015.””
11	What were the criteria to consider a significant association? Did you select variants that were significant in one or other discovery datasets or only from meta-analyses of discovery GWAS studies? Given differences in imputation reference panels for discovery studies (WGS in the Iceland samples and 1000G in the other cohort), are the rare variants present only in one discovery study? Include the number of variants tested in the discovery for each study.	The criterium for significant association is significance in terms of the weighted Bonferroni correction (described on p. 24, lines 542-551) in the Iceland-Denmark meta-analysis. Rare variants may be present only in one discovery study, 35 million variants are tested in Iceland, while 26 million variants are tested in Denmark. In total 40 million variants are tested either in Iceland or Denmark, as stated in Methods, thus $35+26-40 = 21$ million variants are tested in both Iceland and Denmark. These requested numbers have now been added to Methods on page 23, lines 527-530: “35 million variants are tested in Iceland, while 26 million variants are tested in Denmark.” and on page 24, lines 539-541: “In total 40 million variants are tested either in Iceland or Denmark, of which 21 million variants are tested in both datasets. Rare variants may therefore be present only in one discovery study.”
12	Related to above, most of the variants shown in Table 2 are within a gene, which suggest that there is an over-representation of imputed data in coding region in the discovery and/or the results are driven by one study with more dense imputation.	We believe that the question is a misunderstanding and we would like to clarify that there is no overrepresentation of imputed data in coding regions. We wish to highlight that most of our variants are not exons as can be seen in Table 1. Only 6 of the 13 variants are in exons. To clarify, our imputation set is based on whole genome-sequencing, not whole-exome sequencing, and there is no over-representation of imputed data in coding regions (PMID: 25807286). However, we and others have observed an enrichment for associations in coding regions (PMID: 26854916), even though there is no over-representation of imputed data in coding regions.
13	It is not clear how relatedness and population stratification were accounted for in statistical analyses in the Iceland study. Also confusing is the description of thresholds for significance based on variant impact which suggest that the genotypes were predominantly exonic variants.	We apologise for not being clear. A linear mixed model implemented by BOLT-LMM (PMID:25642633) was used to test for association between sequence variants and suPAR levels, and this method accounts for population stratification and cryptic relatedness. Additionally, as stated in Methods (p. 24, lines 534-536), we used linkage disequilibrium (LD) score regression to account for distribution inflation due to cryptic relatedness and population stratification (PMID: 25642630). In Methods we also state that around 214K of the 40M variants are exonic (p.24, lines 545-549), so we are not able to see this from the reviewer’s point of view that the genotypes were

		predominantly exonic variants. To clarify, we have added the following on page 24, line 534-536: “BOLT-LMM accounts for cryptic relatedness and population stratification, and we additionally used linkage disequilibrium (LD) score regression to account for distribution inflation in the dataset due to cryptic relatedness and population stratification (PMID: 25642630).”
Referee #2: molecular epidemiology, GWAS, inflammation		
	This is an interesting GWAS study of an inflammatory marker named su-PAR. To get further insights into the cause of chronic inflammation, it is of highly interest to study the genetics of different inflammatory markers and therefore I think this study is of high importance. The statistical methods are sound, the manuscript reads well.	We thank the reviewer for their positive comments and interest in our manuscript’s findings. We are grateful for their constructive comments and we believe the reviewer’s suggestions have helped to significantly improve our manuscript.
#	Referee	Response to reviewer
1	I miss genome-wide genetic correlation analysis, especially with CRP (2018 paper). I recommend to use ld score regression.	As mentioned above in response to Reviewer 1 Comment #9, we have performed a genetic correlation analysis between suPAR and CRP. The genetic correlation between suPAR and CRP was calculated to be 0.2351 (SD=0.0533, P=1.03e-5), using suPAR from Iceland/Denmark meta-analysis and CRP from UK biobank. This additionally requested analysis has been added to the manuscript. In Methods on page 27, lines 615-620: “The genetic correlation between suPAR and CRP was calculated using suPAR from our Iceland/Denmark meta-analysis and CRP from UK biobank. We additionally investigated overlap of findings between the suPAR-associated variants identified in our study and CRP-associated variants from the GWAS-catalog (https://www.ebi.ac.uk/gwas/). CRP-associated variants reported in the GWAS-catalog were checked for overlap with our suPAR-associated variants + LD-classes (all variants with $r^2 > 0.80$).” In Results, page 14, lines 297-307: We performed a genetic correlation analysis between suPAR and CRP. The genetic correlation between suPAR and CRP was calculated to be 0.2351 (SD=0.0533, P=1.03×10⁻⁵), using suPAR measurements from the Icelandic/Danish meta-analysis and CRP measurements from UK Biobank. We have additionally checked for CRP variants from the GWAS-catalog for our suPAR-associated variants and their LD-classes (all variants with $r^2 > 0.80$). Only the FUT2 variant (rs601338) was found to be an overlapping variant, where the same variant was reported in Han et al (PMID: 30388399). We have additionally

		searched all the genes that we report to see if they have been reported with CRP, of which the only overlap we find in addition to FUT2 is ABO, where there is a variant at chr9:133266942 (rs643434) reported to associate with CRP in the CRP GWAS meta-analysis by Ligthart, et al (PMID: 30388399). This variant has $r^2=0.17$ with the ABO variant found in our study.
2	I miss conditional analysis using GCTA. Are more variants in a single locus associated? This may also improve variance explained.	We performed conditional analysis using Icelandic individual level data, where linkage disequilibrium data is available from the same population. This is in contrast with methods such as GCTA which use summary-level data only. This resulted in the identification of two additional variants in the PLA2R1/LY75 locus: rs114821641 and rs755902185. The other loci did not have additional conditionally significant variants. In the Icelandic data we are able to do a proper conditional analysis since we have the full LD info for the same population, so we believe the approach we did is the best one for our data. If the reviewer insists, we would gladly do the GCTA using the meta-analysis results, however, we do not believe this would be useful. We have now added that conditional analysis has been performed Page 5, lines 111-113: “Two of the 13 genetic variants (rs114821641 and rs755902185 located in the PLA2R1/LY75 locus) were identified via conditional analysis using the Icelandic data exclusively, where linkage disequilibrium data is available from the same population.” Page 24, lines 551-554: “To identify whether several variants in a single locus are independently associated with suPAR, we performed conditional analysis using Icelandic individual level data, where linkage disequilibrium data is available from the same population. This is in contrast with methods such as GCTA which use summary-level data only.”
3	I would recommend the authors to look into potential eQTL effects of the variants found in existing databases. This would provide further info with regards to causal genes.	As requested, we have now looked up all correlated cis-eQTL ($r^2>0.8$). This has been added as Supplementary Table 14, with the sources used listed in Supplementary Table 15. In methods, page 27, lines 610-614: “We tested if our variants were in high LD ($r^2>0.8$) with one more top cis-eQTL based on various tissues and 17 sources including GTEx and Icelandic data (See Supplementary Table 15 for list of sources). For the Icelandic data, RNA sequencing and estimation of association between sequence variants and gene expression have been described in a recent publication (PMID: 32581359).” In Results, page 14, lines 292-295: “We tested if our variants were in high LD ($r^2>0.8$) with one more top cis-eQTL based on various tissues and 17 sources

		including GTEx and Icelandic data. Results are reported in Supplementary Table 14 and sources listed in Supplementary Table 15.”
4	The authors list in the text the numbers that are in the table 1. Please omit this text, the readers can find it in the tables. I would suggest to explain gene functions in a separate box.	We understand that for the reader, the numbers may appear as repetition from table 1. On the other hand we feel that it is difficult not to introduce each variant without stating the variant’s effect, p-value and allele. Our intentions were to make it easier for the reader to find the information instantly without having to refer to the table. We also appreciate the suggestion to explain gene functions in a separate box, however we would prefer to keep the gene functions written under each locus as sentences within the loci subheadings without separating gene functions into a separate box. We note that unnecessary figures or tables are not encouraged by the Communications Biology guidelines when it can be stated briefly in the text instead. Additionally, the BiNGO analysis acts as a form of summary of the gene functions that are overrepresented, where figure 2 highlights the overrepresented gene functions in an alternative and condensed format in contrast to a gene-by-gene gene functions box. We therefore hope that the reviewer understands our considerations for the most appropriate way to report our results, but if the reviewer insists, we can remove numbers and create a gene function table.
5	I would also like to see proper mendelian randomization analysis for the identified PHEWAS findings.	As requested by the reviewer, we have performed mendelian randomization analysis for the identified phewas findings. No significant findings were found. We have added the following to the manuscript: Results, Page 14, lines 287-290: “We performed mendelian randomization analyses for suPAR and the identified pheWAS findings using phenotypes available in the Icelandic population cohort. No significant results were found (Supplementary Table 13).” Methods, Page 26, lines 601-608: “We performed mendelian randomization analyses for suPAR and the identified pheWAS findings using phenotypes available in the Icelandic population. The analyses were performed using the R Package “MendelianRandomization”. Several methods were used, including Simple median, Weighted median, Penalized weighted median, Inverse-variance weighted (IVW), Penalized IVW, Robust IVW, Penalized robust IVW, MR-Egger, Penalized MR-Egger, Robust MR-Egger, and Penalized robust MR-Egger. Mendelian randomization for suPAR and 10 phenotypes were performed, including type 1 diabetes, autoimmune diseases as a general category, obesity, rheumatoid arthritis, B12, fasting glucose, alkaline phosphatase and potassium.”
6	Methods: Which software was used for the GWAS meta-analysis?	Variants in the Icelandic and Danish cohorts were imputed using software developed at deCODE genetics based on the

	Please mention.	IMPUTE HMM model [PMID: 17572673] as previously described [PMID: 25977816]. A linear mixed model implemented by BOLT-LMM [PMID:25642633] was used to test for association between sequence variants and suPAR levels. We used publicly available software (URLs listed below) in conjunction with the above described algorithms in the sequencing processing pipeline (Whole-genome sequencing, Association testing, RNA-seq mapping and analysis): BWA 0.7.10 mem, https://github.com/lh3/bwa GenomeAnalysisTKLite 2.3.9, https://github.com/broadgsa/gatk/ Picard tools 1.117, https://broadinstitute.github.io/picard/ SAMtools 1.3, http://samtools.github.io/ Bedtools v2.25.0-76-g5e7c696z, https://github.com/arkq5x/bedtools2/ Variant Effect Predictor https://github.com/Ensembl/ensembl-vep BOLT-LMM https://data.broadinstitute.org/alkesgroup/BOLT-LMM/downloads/ IMPUTE2 v2.3.1 https://mathgen.stats.ox.ac.uk/impute/impute_v2.html dbSNP v140; http://www.ncbi.nlm.nih.gov/SNP/ BiNGO v3.0.3 https://www.psb.ugent.be/cbd/papers/BiNGO/Download.html Cytoscape v3.7.1 https://cytoscape.org/download.html We used R extensively to analyze data and create plots. The above text has been added in Methods under the heading “Code Availability” on page 28, starting on line 637. Additionally we have added the following line on page 23, line 527-529: “A linear mixed model implemented by BOLT-LMM (PMID:25642633), was used to test for association between sequence variants and suPAR levels, assuming an additive genetic model.”
7	Danish cohort methods: random selection patients for su-PAR measurements?	We performed suPAR measurements on 14,367 consecutive participants that were included in the Danish Blood Donor Study (consecutive sampling). We have now made this clear on page 19, lines 414-416: “[...] suPAR was measured in 14,367 consecutive DBDS participants from March 1st 2010 until December 10th 2010 of which 12,177 (84.8%) participated in the GWAS after fulfilling quality control requirements.”
8	E-Risk study methods: first sentence about Dunedin study.	Thank you for noticing this error. This has been corrected. Page 20, lines 440-441: “Environmental Risk (E-Risk) Longitudinal Twin Study: suPAR levels were measured at age 18 years in the E-Risk Study, as previously described.”

Referee #3: statistical genetics, GWAS		
	The authors have presented an in-depth study, investigating the potential genetic role in the variation of suPAR levels as well as its potential role in relation to risk of disease, such as inflammation. suPAR, which is normally used as a non-specific biomarker for chronic inflammation, have been extensively studied in relation to lifestyle factors, however the genetics behind changes in marker is lesser known. The authors here present two independent GWAS in an Icelandic and a Danish cohort, which were then meta-analysed together. Additionally, they validate their results in a UK-based cohort as well as a New Zealand-based. They present possible candidate genes based on genomic location and gene ontology / biological processes analysis. Lastly a heritability analysis is performed as well as a pheWAS for computed polygenic risk scores as well as for the independently associated variants from the meta-analysis. This is a well-written article, and it contributed to further knowledge within the field. However, there are some points that could improve even more with clarification or further discussion/commenting.	We thank the reviewer for their thorough review of our manuscript and constructive feedback.
#	Referee 3	Response to reviewer
1	It is mentioned in the introduction that life factors associated with suPAR have been extensively studied, especially smoking. In the main analyses, only sex is used as a covariate. Do the authors have access to lifestyle factors such as smoking status? Would this and other potential life style factors affect the outcome of the primary results in the GWASs? Is there a reason principal component were only used in the replication analyses?	We would like to highlight that both sex and age were used as covariates. For the Icelandic data, information on smoking status was available for 30,469 of the 35,559 individuals with suPAR measurements, including 11,093 non-smokers and 19,376 smokers, where “smoker” was defined as “ever smoker”. Smokers had higher suPAR levels than non-smokers (Effect=0.12 SD, P=2.2e-23 from t-test). To assess if the including of smoking as a covariate could have an effect on the GWAS outcomes, we performed two GWAS: 1) the 30,469 individuals with available information on smoking status, unadjusted for smoking; and 2) the same 30,469 individuals, adjusted for smoking. A test of difference in the GWAS results from 1) and 2) revealed no differences, with heterogeneity p-

		values ranging from 0.94 to 1.00 (supplementary table 2). This additional analysis has been added to the manuscript: In results page 7 lines 130-137: “As suPAR levels have strong associations with smoking, we investigated whether smoking status would affect the outcome of the suPAR GWAS results. Using the Icelandic cohort, we performed two GWASs; the 30,469 individuals with available information on smoking status, unadjusted for smoking; and the same 30,469 individuals, adjusted for smoking. Smokers had higher suPAR levels than non-smokers (Effect=0.12 SD, $P=2.2 \times 10^{-23}$ from t-test). A test of difference in the GWAS results between the two above-mentioned Icelandic GWASs revealed no difference when adjusting for smoking, with heterogeneity p-values ranging from 0.94 to 1.00 (Supplementary Table 2).” In Methods, page 25, lines 566-573: “As suPAR levels have strong associations with smoking, we investigated whether smoking status would affect the outcome of the suPAR GWAS results. Smoking status was available for 30,469 of the 35,559 individuals with suPAR measurements in the Icelandic data. This includes 11,093 non-smokers and 19,376 smokers, where “smoker” was defined as “ever smoker”. To assess if the inclusion of smoking as a covariate could have an effect on the GWAS outcomes, we performed two GWASs: 1) the 30,469 individuals with available information on smoking status, unadjusted for smoking; and 2) the same 30,469 individuals, adjusted for smoking. A test of difference (heterogeneity) in the GWAS results was performed.”
2	How do the heritability estimates look in the other non-twin-based cohorts? Is there a possibility to calculate SNP heritability in the Icelandic and Danish cohort? Would it differ much from the PRSs, i.e. explain more of the difference between the variance explained by the PRS and the genetic additive effects explained in the MZ twin setting? Would this explain some of the gap between the MZ estimates and the PRSs?	SNP heritability was calculated to be 0.1252 (SD: 0.0487) based on the Icelandic cohort. We estimated the narrow sense heritability of phenotypes with ld score regression (PMID: 25642630), using an LD score map calculated with high quality markers from the Icelandic population.
3	When talking about the results and discussing further downstream analyses, only the variants from the meta-analysis considered. As of now, it is a bit unclear in the text whether the 11 loci are completely overlapping between the Icelandic and the Danish	We understand that we could present the differences between the two GWAS results more clearly. Currently in the main text we explain how the variants were tested for heterogeneity between the two cohorts and how only the variant in the ST3GAL6 locus shows evidence of heterogeneity, however the direction of effects are consistent between the two cohorts and is significant in both cohorts. In the case of the two very rare variants in the PLA2R1/LY75 locus: rs114821641 and

	cohort or if there were cohort-specific signals. It is clear in S Table 1, the cohort section, that rs755902185 was not picked up in the DBDS for example, but I think it would be neat to present it also in the main text for clarification.	rs755902185, it is correct that these were not picked up in the DBDS cohort. We identified these two rare variants by performing conditional analysis using the Icelandic data, where linkage disequilibrium data is available for the same population. This resulted in the identification of the two variants. We have now made this more clear. Page 5, line 111-113: “Two of the 13 genetic variants (rs114821641 and rs755902185 located in the PLA2R1/LY75 locus) were identified via conditional analysis using the Icelandic data exclusively, where linkage disequilibrium data is available for the same population.”
4	The suPAR measurements have been measured at different time points in the cohorts (at 18yrs in E-Risk and 38yrs in Dunedin and not stated in the rest). It is stated that suPAR is stable in terms of minimal circadian changes, but does the time of measurement affect further analysis? Does the age component in the regression capture any possible variation due to age (if present)?	As is referenced previously, suPAR is known to be extremely stable with respect to circadian rhythm changes but also with respect to pre-processing storage of whole blood, storage time and repeated freezing-thawing cycles. Therefore we do not believe that the time of measurement affects further analysis. To prevent evaporation of water in long-term storage of our blood samples in DBDS, Dunedin and E-Risk, samples are kept at -80 degrees Celsius. However, we cannot exclude that time in freezer can have some influence on protein concentration. If this is the case, it should weaken rather than strengthen our results. Three studies (as referenced in the manuscript ref 27-29) have investigated the minimal circadian variation of suPAR, and none of them found any influence on suPAR levels. Another study found that suPAR is stable (in contrast to CRP) even during primary percutaneous intervention in patients with ST-segment elevation myocardial infarction (PMID: 25305537). With respect to suPAR and age, we have previously seen that suPAR increases with higher age in Danish blood donors (PMID: 25329298) and with age adjustment included in all our models, we correct for this. The fact that we have included suPAR measurements from cohorts of such a great age span is inferred as a strength rather than a limitation of our study. We would also like to note that in the Dunedin cohort, a recent study showed that plasma suPAR levels were correlated across 7 years, at age 38 and 45 $r = 0.58$, $p < 0.0001$ (PMID: 32766674). For comparison, CRP levels were correlated at age 38 and 45, $r = 0.26$, $p < .001$. This was also observed in a study by Haupt et al. from 2019 (PMID: 30679937) where they found that suPAR is strongly correlated ($r=0.55$) in individuals taken 5 years apart. These studies indicate that individuals tend to retain their rank in the population on suPAR as they age and further support suPAR’s reputation as a stable chronic inflammation marker. Page 4, lines 71-72: “Additionally, unlike CRP, suPAR is a stable biomarker as circadian changes in plasma suPAR are minimal, and suPAR

		measurements in individuals have been shown to be correlated across five and seven years (PMID: 30679937, PMID: 32766674)".
5	Figure 1: The Manhattan plot is a bit confusing. The labelled variants are the independently associated variants at the 11 loci, however they are rarely the lead variant at respective locus (i.e. lowest p value). How are the independent variants at each locus defined?	We understand that the Manhattan plot may appear confusing due to the independent lead variant at each locus not always being the variant with the lowest P-value. Regarding the PLAUR locus, we have identified an error in the Manhattan plot which plots some variants that were removed at an earlier stage of quality control. They have now been removed, and therefore the only remaining variant in the PLAUR locus with a lower p-value than the lead variant rs36229204 is rs2302524, which, as described on page 11 lines 226-2233, was independently significant in both the Danish and Icelandic GWASs, but with opposite effect directions and therefore was not included in Table 1 as the 14th signal. The other loci where the variant with the lowest p-value and the labelled lead variant differ can be explained by how we select the lead variant based on impact-group-corrected p-value as described page 5, lines 99-105, as well as in Methods. Figure 1 has been updated with clearer annotations of the 11 loci including: larger font size, annotation labels spaced out to avoid overlap on plotted points, lines indicating the designated variant's label in "dense" regions of the plot, and the "14th signal" rs2302524 which has now been annotated but shaded grey to signify it is not included as one of the 13 included variants from Table 1:  Figure 1: suPAR GWAS Meta-Analysis Manhattan plot (N=47,736), showing the 11 genome-wide significant loci and the 13 independently significant variants associated with suPAR (red points). The negative log₁₀ transformed P values for variants are plotted by chromosomal location. Y axis begins at $P=1 \times 10^{-5}$.
6	How many phenotypes are overlapping between the Icelandic cohort and in UK Biobank? Is there a possible reason why the UK Biobank based pheWAS does not yield as many as or similar results as the Icelandic based?	Unfortunately we are unable to provide simple answers to these questions. We wish to highlight that the Icelandic cohort and the UK biobank are two separate cohorts that make comparisons difficult. The phenotypes collected in the Icelandic population cohort predominantly originate from healthcare facilities and participants were recruited for specific research projects, whereas the UK Biobank population are overall healthy volunteers (5.5% response rate to recruitment). Additionally there are different case/control sizes for the different phenotypes in each cohort, with the possibility of the specific phenotype being measured/determined using alternative methods. Finally there are phenotypes that only

		appear in one of the cohorts and not the other. These are examples of how making a comparison between the results from the two cohorts would be challenging and may explain why the UK Biobank pheWAS does not yield similar results to the Icelandic cohort.
7	In general, I think it would be helpful to add footnotes to the tables, as with S. Table 4, making them even more clear to follow; e.g., adding a footnote for nAff and nCon in S. Table 6, which I assume to be number of affected and number of controls respectively.	Thank you for this suggestion. We have now added footnotes for some abbreviations that have been used in the tables to avoid possible confusion.
8	For better comparison, it would be good if the allele frequencies (effect and minor) are stated as a fraction rather than a percentage in the supplementary tables (S Table 1-2,8-11) to match the tables in the main article or vice versa.	We agree with the reviewer's suggestion. We have therefore updated all allele frequency columns in all relevant supplementary tables to show all allele frequencies in fraction format.
9	Line 235 is a bit confusing, "to investigate the combined effect of these suPAR-associated genetic variations", assuming it is the combined effect of all significantly associated variants in the GWASs. The sentence might benefit from being rephrased.	Thank you for informing of us about this sentence's potential for being misunderstood. The method of calculating the suPAR-PRSs is described in the methods of which we state that 630,000 variants are used and not only the 13 significantly associated variants. We have simplified the sentence by removing the word "these" from the sentence. Page 13, lines 263-264: "To investigate whether the combined effect of suPAR-associated genetic variations were associated with specific phenotypes, we performed pheWASs using suPAR polygenic risk scores (PRSs) as the exposure."
10	Line 374-375: "Environmental Risk (E-Risk) Longitudinal Twin Study: suPAR levels were measured at age 18 in the Dunedin study as previously described": I assume Dunedin is wrongly stated and should be E-Risk?	Thank you for noticing this error. As mentioned in Reviewer 2 Comment #8, this has now been corrected. Page 20, lines 440-441: "Environmental Risk (E-Risk) Longitudinal Twin Study: suPAR levels were measured at age 18 years in the E-Risk Study, as previously described."
11	What software was used for the heritability and GWAS analyses? And why different software for all different analyses?	For the heritability analysis we used Mplus Version 7.4 (Muthen & Muthen, Los Angeles, CA) . Association testing was performed using a linear mixed model implemented by BOLT-LMM. See answer to Reviewer 2, Comment #6. We are unsure of what the reviewer wishes to ask us in "why different software for all different analyses?". We are unaware of a single reliable program that is able to perform all analyses featured in our study. We wish to clarify that we use the same software for both GWASs and the meta-analysis, and we use the same software for both validations. We have updated the Methods section as follows: Page 23, line 524:

		“We used Mplus Version 7.4 (Muthen & Muthen, Los Angeles, CA) for the analysis.” (heritability analysis). Page 23, lines 527-530: “A linear mixed model implemented by BOLT-LMM, was used to test for association between sequence variants and suPAR levels, assuming an additive genetic model.” Additionally software has been listed in Methods under the heading “Code Availability” on page 28, starting on line 637
12	Line 89-92: Are the additive effects + shared environment + nonshared environment supposed to add up to 100%?	We used Mplus Version 7.4 for the heritability analyses, and yes, they are supposed to add up to 100%. We have double-checked and the numbers are correct as they were reported (60.1%, 10.3%, 30.4%). The numbers outputted in Mplus were “A 0.601, C 0.103, E 0.304”. To avoid confusion, we have now rounded them down to 60%, 10%, and 30%, including the confidence intervals. Page 5, lines 92-96: “Using a univariate twin model, we found that additive genetic effects accounted for 60% (95% CI: 38%–82%) of the variation in suPAR levels, while shared environmental influences accounted for 10% (95% CI: 0%–31%) of the variance and nonshared environmental influences accounted for 30% (95% CI: 26%–35%) of the variance in suPAR levels.”

Reviewers' comments:

Reviewer #1 (Remarks to the Author):

The authors answered all my questions. I have no further comments.

Reviewer #2 (Remarks to the Author):

The authors have sufficiently addressed my concerns and have added the analyses that I asked for. This has improved the manuscript. Well done.

Reviewer #3 (Remarks to the Author):

The authors have made a great work reworking the manuscripts following the recommendations and suggestions by the reviewers. The vast majority of my questions and comments have been sufficiently answered, especially in combination with the other reviewers and the addition of the code availability section, which has improved the manuscript significantly.

However, with regards to the further revisions I still have some minor comments.

If the statement regarding the missing heritability should be left in the manuscript, I do think the comparison should be made to the SNP heritability rather than the PRS as that is a fairer comparison with regards to heritability.

I would like a small clarification on why the replication analyses were performed in SNPAssoc with 10 PC rather than BOLT-LMM but I am assuming it is due to the lack of LD information in the replication cohorts and thus not being able to account for relatedness and population stratification using BOLT-LMM?

However, if the authors have access to all raw data for all cohorts, wouldn't it be possible to calculate LD within the populations additional to the Icelandic cohort where it was already available? If deemed necessary/useful, then the conditional analysis could have been extended to include Denmark as well.

If the mendelian randomization added as a response to the request by another reviewer should be included, I do believe the results need a short discussion/explanation even if non-significant, as a complement to the pheWAS. I assume the IVs are the lead variants, and there seem to be some quite large outliers in some MR analyses.

Lastly, there are quite a few typographical errors (lacking spaces, lacking interpunctuation etc.) where revisions have been made, so just make sure that the text is carefully looked through in the proofs.

Response to Reviewers for the suPAR GWAS Meta-analysis Manuscript

We thank Reviewer #1 and Reviewer #2 for their review of our manuscript.

Referee #3: GWAS, nephrology

	The authors have made a great work reworking the manuscripts following the recommendations and suggestions by the reviewers. The vast majority of my questions and comments have been sufficiently answered, especially in combination with the other reviewers and the addition of the code availability section, which has improved the manuscript significantly. However, with regards to the further revisions I still have some minor comments.	We would like to thank the reviewer once again for their valuable comments. We have responded to the minor comments that the reviewer has raised in a point-by-point manner below.
#	Comment	Response to reviewer
1	If the statement regarding the missing heritability should be left in the manuscript, I do think the comparison should be made to the SNP heritability rather than the PRS as that is a fairer comparison with regards to heritability.	We agree with the reviewer. We have therefore added SNP heritability in the Icelandic population to the methods section, the estimate of the SNP heritability under the heritability subsection of the results section, and replaced the statement in the heritability discussion with the SNP heritability estimate. In Methods, page 23, lines 515-518: “We additionally calculated SNP heritability based on the Icelandic cohort. We estimated the narrow sense heritability of suPAR with LD score regression (PMID: 25642630), using an LD score map calculated with high quality markers from the Icelandic population.” In Results, page 5, lines 96-97: “We additionally calculated the SNP-based heritability based on the general Icelandic population cohort. The SNP heritability estimate was calculated to be 12.5% (SD: 4.9%).” In Discussion, page 18, lines 379-384: “However, it must be noted that the SNP-based heritability based on the Icelandic cohort was calculated to be 12.5% (SD: 4.9%). As our univariate twin model estimated that additive genetic effects account for 60% of the variation in suPAR levels, we are unable to account for much of the heritability of suPAR, a situation frequently reported in GWASs of other phenotypes.” Code Availability, page 29, line 649: “LD Score Regression software; https://github.com/bulik/ldsc”

2	I would like a small clarification on why the replication analyses were performed in SNPAssoc with 10 PC rather than BOLT-LMM but I am assuming it is due to the lack of LD information in the replication cohorts and thus not being able to account for relatedness and population stratification using BOLT-LMM?	We would like to clarify why the replication analyses were performed in SNPAssoc rather than BOLT-LMM. To be clear, the replication analyses using SNPAssoc were performed on only a small number of top-hit variants from the GWAS meta-analysis. They do not constitute a replication GWAS. Therefore, the analyses are a simple test of an a priori hypothesis and do not warrant a more complicated analysis approach.
3	However, if the authors have access to all raw data for all cohorts, wouldn't it be possible to calculate LD within the populations additional to the Icelandic cohort where it was already available? If deemed necessary/useful, then the conditional analysis could have been extended to include Denmark as well.	Though it is technically possible to calculate LD within the Danish population, we do not believe this will be useful as the LD will be very similar in Denmark and Iceland and will therefore not change the results. We wish to underline that the main use of the conditional analysis in the paper is to check that the three variants at the PLA2R1/LY75 locus are uncorrelated. After receiving this comment, we additionally verified that these three variants are also uncorrelated in the Danish imputation. To reiterate, the conditional analysis was used to determine that the three PLA2R1/LY75 variants are uncorrelated, and we do not believe that it would be useful to rerun the entire analysis with conditional analysis including Denmark.
4	If the mendelian randomization added as a response to the request by another reviewer should be included, I do believe the results need a short discussion/explanation even if non-significant, as a complement to the pheWAS. I assume the IVs are the lead variants, and there seem to be some quite large outliers in some MR analyses.	As the mendelian randomization analysis was requested by another reviewer, we feel obligated to include the analysis. However, we agree that the way we have presented the analysis can be simplified and improved. We have therefore removed all MR results that are not IVW and MR-Egger (with the intercept). We have additionally added the MR-PRESSO global test (PMID: 29686387) to detect outliers and remove them. These results have been added to Supplementary Table 13, and in the analyses where outliers were found, the IVW analysis was performed again with outliers removed. The IVs are indeed the lead variants, i.e. the variants we report significant for suPAR. This has been made more clear in our methods section. We would like to note that our MR analyses are underpowered and although some outliers were detected for some phenotypes, the removal of these does not improve the significance of the MR analyses. Therefore we do not believe it would be useful to provide a short discussion/explanation on the MR analyses. We believe there are more interesting results in the paper that have been prioritized to be considered as part of the discussion section. In Methods, pages 26-27, lines 595-603: “The analyses were performed using the R Package “MendelianRandomization”, using the inverse variance weighted (IVW) and MR-Egger methods. The MR-PRESSO global test (PMID: 29686387) was additionally used to detect possible outliers and remove them. In the instances where

		outliers were found, the outliers were removed and the IVW method was reutilized. . The instrumental variables (IV's) used are the variants we report significant for suPAR, i.e. the lead variants. Mendelian randomization for suPAR and 10 phenotypes were performed, including type 1 diabetes, autoimmune diseases as a general category, obesity, rheumatoid arthritis, B12, fasting glucose, alkaline phosphatase and potassium.” In Results, page 14, lines 279-280: “‘No significant results were found and the removal of outliers detected using MR-PRESSO did not improve the estimates significantly (Supplementary Table 13).” In Supplementary Table 13: Supplementary Table 13 has been simplified by removing all MR analyses other than IVW and MR-Egger (with the intercept). We have additionally added the MR-PRESSO results and IVW after the removal of outliers.
5	Lastly, there are quite a few typographical errors (lacking spaces, lacking interpunctuation etc.) where revisions have been made, so just make sure that the text is carefully looked through in the proofs.	We apologize for this. We have read through the text thoroughly to ensure that typographical errors that we have identified have been corrected.